# Limitations of acyclovir and identification of potent HSV antivirals using 3D bioprinted human skin equivalents

S. Tori Ellison[1,7], Ian Hayman[2,7], Kristy Derr[1], Paige Derr[1], Shayne Frebert[1], Zina Itkin[1], Min Shen[1], Anthony Jones[2], Wendy Olson[2], Lawrence Corey [2,3,4], Anna Wald [2,3,4,5], Christine Johnston [3,5], Youyi Fong [4], Marc Ferrer [1] ✉ & Jia Zhu [2,4,6] ✉

Herpes simplex virus (HSV) infection poses global public health concerns with lifelong impacts. Acyclovir, the standard therapy, has limited efficacy in preventing subclinical shedding, and drug resistance occurs in immunocompromised patients, highlighting the need for novel therapeutics. Here we show that acyclovir is significantly less effective in skin-derived keratinocytes than donor-matched fibroblasts. Using 3D bioprinted human skin equivalents (HSEs) in a 96-well plate format, we have screened 738 compounds with broad targets and mechanisms of action, identifying potent antivirals, including 23 known or experimental HSV treatments. Unlike acyclovir, antivirals against HSV helicase/primase or host replication pathways display similar potency across cell types and donor sources in both 2D and 3D models. The reduced potency in keratinocytes may explain acyclovir's limited clinical efficacy. Our 3D bioprinted HSE assay platform enables the integration of patient-derived cells early in drug development and offers a physiologically relevant approach for HSV drug discovery.

Herpes simplex virus type 1 (HSV-1) and type 2 (HSV-2) cause recurrent oral and genital ulcer diseases and infect two-thirds of the global population[1,2]. Both HSV-1 and HSV-2 can cause neonatal herpes with high mortality and devastating neurological impairment[3,4]. HSV-2 infection increases the risk of human immunodeficiency virus (HIV) acquisition[5–7], with a recent study estimating 420,000 of 1.4 million newly acquired HIV infections annually attributable to HSV-2 infection[8]. Despite successes in animal studies, candidate HSV-2 vaccines faltered in human trials[9–11]. Acyclovir, the primary treatment for HSV infection, alleviates disease severity and shortens outbreak duration[12] but faces drug resistance in immunocompromised patients[13,14], has limited efficacy in preventing subclinical reactivation[15],

and fails to address the increased risk of HIV transmission[16,17]. Thus, novel therapeutic strategies and antiviral agents are needed to address the public health burden of HSV infection.

Drug development relies heavily on 2D cell monolayers and animal models to assess drug safety and efficacy. Conventional in vitro cultured cell monolayers facilitate rapid and robust infectivity assays but do not recapitulate complex cell-cell and cell-matrix interactions reflective of native tissues[18,19]. Consequently, these cellular assays often fail to predict drug responses in humans accurately[20]. Animal models lack the predictive value and biological relevance to humans for effective drug discovery and development[21]. This dilemma is evident in drug candidates failing to progress from Phase I to Phase III clinical

[1]Department of Preclinical Innovation, National Center for Advancing Translational Sciences, National Institutes of Health, Rockville, MD, USA. [2]Department of Laboratory Medicine and Pathology, University of Washington School of Medicine, Seattle, WA, USA. [3]Department of Medicine, University of Washington School of Medicine, Seattle, WA, USA. [4]Vaccine and Infectious Disease Division, Fred Hutchinson Cancer Center, Seattle, WA, USA. [5]Department of Global Health, University of Washington School of Medicine, Seattle, WA, USA. [6]Institute of Stem Cell and Regenerative Medicine, University of Washington, Seattle, WA, USA. [7]These authors contributed equally: S. Tori Ellison, Ian Hayman. ✉e-mail: marc.ferrer@nih.gov; jiazhu@uw.edu

trials, resulting in continued high costs for developing new pharmaceuticals[22,23]. 3D in vitro models closely mimicking human tissues and organs offer opportunities to circumvent the limitations in clinical predictability of current drug R&D platforms[24,25].

Bioengineered tissues bridge the gap between existing cellular and animal models and the complexity of human hosts and offer a powerful platform for expediting drug discovery, toxicity screening, and preclinical testing in a high-throughput, cost-effective manner. These in vitro human tissue models are designed to replicate key features of in vivo organs, such as cell type composition, 3D microarchitecture, functional tissue interfaces, and organ-specific microenvironments[26]. They offer unique opportunities for real-time visualization and high-resolution analysis of biological processes often unattainable in simpler cellular and animal models.

Skin and mucosal barriers are the primary tissues of concern for HSV initial infection and recurrent viral replication following reactivation. Skin tissue comprises the epidermis, dermis, and subcutaneous layer, forming a physical barrier protecting against pathogens and harmful environmental substances[27]. The epidermis primarily consists of keratinocytes, which can differentiate into stratified layers, from the innermost stratum basale to the outmost stratum corneum. Human biopsy studies of recurrent HSV infection indicate basal keratinocytes at the dermal-epidermal junction (DEJ) serve as the replication targets for HSV reactivation through sensory nerve innervation and neurite extension into DEJ[28]. Our recently developed 3D skin-on-chip also demonstrated that basal keratinocytes were the most susceptible targets for HSV infection during keratinocyte differentiation[29]. Thus, both in vivo and in vitro evidence highlight the importance of using multicellular tissue models that recapitulate skin architecture when developing HSV therapeutics.

Here we develop bioprinted human skin equivalents (HSE) that differentiate into a stratified epidermis at the air-liquid interface (ALI)[30,31]. We implement a high-throughput screen (HTS) using 3D bioprinted HSE in a 96-well format, leveraging a HSV-GFP reporter virus and high-content imaging. Our 3D bioprinted assay platform identifies potent antiviral compounds, including 23 known anti-HSV candidates. We further investigate the activity of 11 top candidate compounds in adult human skin-derived keratinocytes and fibroblasts, uncovering pharmacological properties distinct among cellular types in 2D and 3D. This work highlights the importance of applying physiologically relevant assays in drug discovery and development.

## Results

### Assessment of HSV susceptibility and acyclovir antiviral potency across different cell types

Herpes simplex virus enters the human body through the skin and mucosa surface and establishes latent infection for the lifetime of the host[32]. The virus reactivates periodically and causes recurrent disease back to the periphery. Basal keratinocytes are the primary cells encountered by HSV reactivation in the skin[28,33]. However, current drug discovery practice utilizes Vero cells and fibroblast cultures for antiviral drug development[34]. To investigate if acyclovir displayed similar effectiveness in keratinocytes, fibroblasts, and Vero cells, we isolated keratinocytes and fibroblasts from six donors (Supplementary Table 1) using 3 mm skin punch biopsies (Fig. 1A). Donor-specific keratinocytes and dermal fibroblasts were separated and matched for comparison. Real-time monitoring of recombinant strain HSV-1 K26 infection via GFP fluorescence (Supplementary Fig. 1) revealed GFP expression peaked at 20, 48, and 36 h post-infection (HPI) in keratinocytes, fibroblasts, and Vero cells, respectively (Fig. 1B). Time to initial GFP detection was significantly earlier in keratinocytes (6.0 HPI) than in fibroblasts (7.9 HPI) ($P < 0.001$) (Fig. 1C). GFP expression increased much faster in keratinocytes compared to both fibroblasts and Vero cells ($P < 0.01$), the maximum GFP doubling rate (mean ± sd) was $2.23 \pm 0.41$, $1.37 \pm 0.39$, and $1.00 \pm 0.15$ per hour in keratinocytes,

fibroblasts, and Vero cells, respectively (Fig. 1D). These results indicated that HSV infection and viral gene expression occurred more rapidly in keratinocytes than in donor-matched fibroblasts or Vero cells.

We then evaluated the potency and efficacy of acyclovir in dose response using donor-matched keratinocytes and fibroblasts. The dose-response curves were established using the peak infection time previously determined in each cell type (Fig. 1B, E–G). Acyclovir exhibited significantly reduced potency in keratinocytes compared to fibroblasts across all six donors (Fig. 1E–G, $P < 0.001$). No donor-specific response was observed to acyclovir treatment. Instead, acyclovir antiviral activities differed dramatically between keratinocytes and fibroblasts across multiple timepoints (Supplementary Fig. 2). The concentration of acyclovir required to inhibit 50% of virus-encoded GFP expression ($IC_{50}$) for each donor was, on average (mean ± sd), 196.7-fold higher in keratinocytes ($67.7 \pm 18.2$ μM) than in fibroblasts ($0.40 \pm 0.2$ μM) and 60.2-fold higher than in Vero cells ($1.14 \pm 0.2$ μM). The cytotoxic concentration required to kill 50% of target cells ($CC_{50}$) was over 600 μM in donor-derived keratinocytes and fibroblasts (Supplementary Fig. 3), indicating the difference in antiviral potency was not due to cytotoxicity. The $IC_{50}$ of acyclovir in keratinocytes was also over twofold higher than published peak serum levels in patients following treatment with three times daily 1000 mg valacyclovir[35]. This suggests that current acyclovir treatment might not be optimal for inhibiting HSV gene expression in the skin epidermis. These results confirmed that antiviral drug potency can vary substantially among cell types tested, emphasizing the importance of employing physiologically relevant cells in antiviral drug testing.

### Development of HSV infection assays on 3D bioprinted human skin equivalents

The significant differences in acyclovir responses between keratinocytes and fibroblasts led us to develop multicellular tissues for early drug discovery that better recapitulate skin in vivo to investigate antiviral potency and efficacy. To create physiologically relevant in vitro human skin models of HSV-1 infection for antiviral screening, we 3D bioprinted human skin equivalents (HSE) in a 96-transwell plate format. The hydrogel containing gelatin, collagen, fibrin, and neonatal human fibroblasts was bioprinted onto the apical side of a 96 transwell insert using a plunger-based 3D bioprinter (Fig. 2A). The dermal tissues underwent 7 days of maturation before neonatal human keratinocytes were seeded on top and submerged in epidermalization media for 7 more days. To induce keratinocyte differentiation and epidermis stratification, the tissues were lifted to air-liquid interface (ALI) using a custom 3D printed adaptor (Supplementary Fig. 4) and cultured in cornification media at the basal surface of the dermis for 7 days (Fig. 2A)[30,31,36]. Hematoxylin and eosin (H&E) staining confirmed that ALI cultures produced HSE with a dermis and differentiated epidermis, including a stratum basale, stratum spinosum, stratum granulosum, and stratum corneum (Fig. 2B Top). Immunohistochemistry (IHC) staining demonstrated cytokeratin 10 (K10) and cytokeratin 14 (K14) expression in the suprabasal and basal layer, respectively, and loricrin and filaggrin in the stratum granulosum and stratum corneum, respectively (Fig. 2B Bottom), indicating properly differentiated epidermis resembling in vivo human skin architecture[37]. The antiviral screen was completed using two fluorescent markers; GFP-fused HSV-1 strain K26[38] was employed to measure virus infection, while fibroblasts were transduced to constitutively express tdTomato to monitor compound cytotoxicity. The infection conditions were optimized by imaging the tissues using a high-content microscope at 24, 48, and 72 h post-infection (HPI). The total fluorescence signal of GFP and tdTomato was quantitated using a maximum projection at a multiplicity of infection (MOI) of 0.1, 1.0, and 10 PFU/cell (Fig. 2C). GFP expression had the most significant difference between mock and infected using an MOI of 0.1 at 48HPI (difference in relative fluorescence units

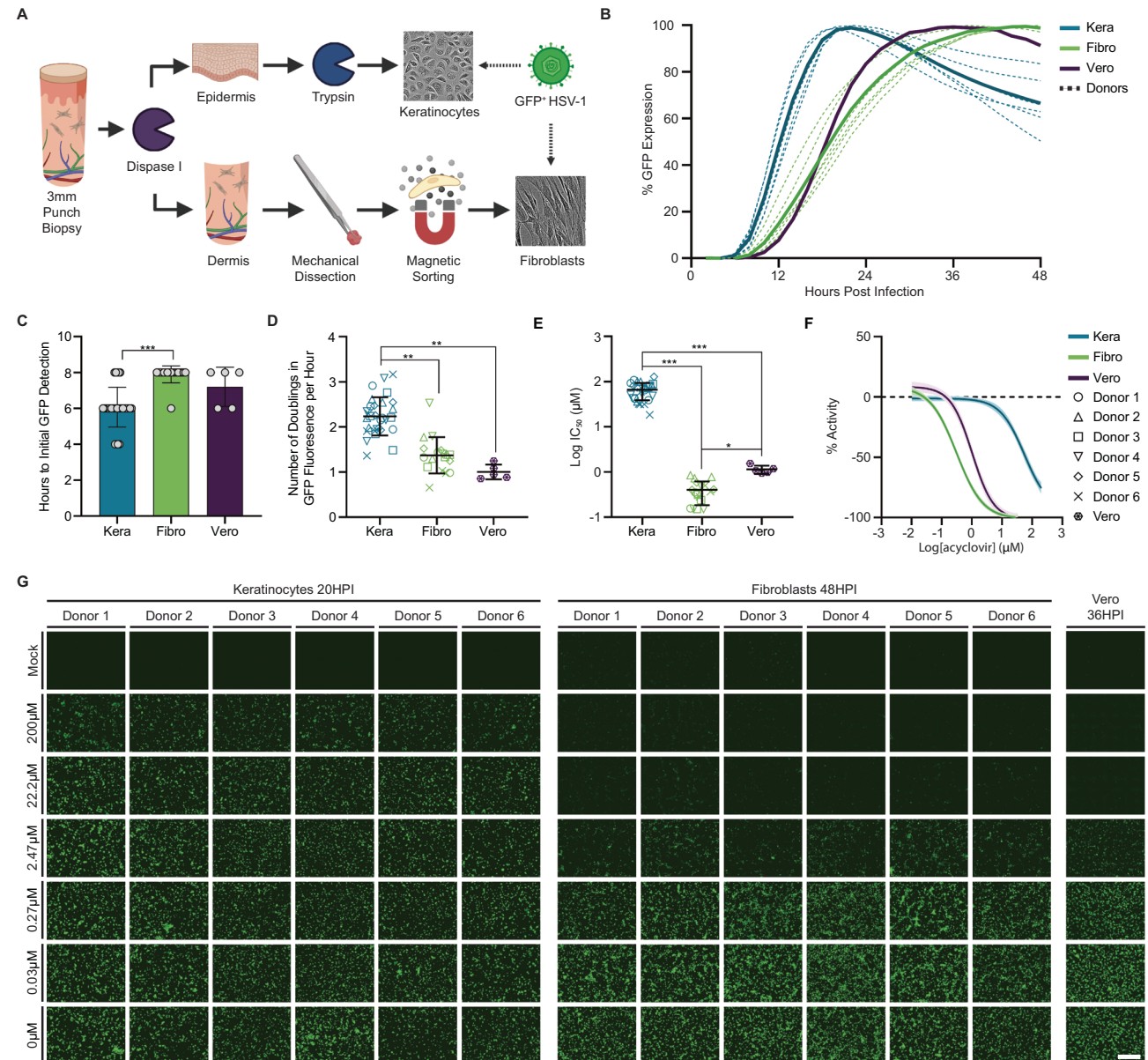

**Fig. 1 | Acyclovir potency in donor-derived primary keratinocytes and fibroblasts and in Vero cells. A** Punch biopsies from six donors were collected and dissociated by enzymatic and mechanical processes (Created in BioRender. Hayman, I. (2025) https://BioRender.com/fmeng2t). **B** Live-cell images of HSV-1-GFP infected Vero cells (purple), keratinocytes (blue), and fibroblasts (green) were taken every two hours. Dotted lines represent individual donors, solid lines represent averages ($N = 3$). **C** Time until HSV-encoded GFP was first detected in each cell type. (*** $P < 0.001$, two-tailed unpaired $T$ test, df = 42.47, keratinocytes ($N = 30$), fibroblasts ($N = 18$), and Vero cells ($N = 5$). **D** Doubling time for virus-encoded GFP fluorescence in 2D monoculture; a value of 1 means one doubling per hour. All biological replicates are plotted by donor. Statistics were determined by five (keratinocytes) or three (fibroblast) averaged biological replicates in six donors. Five biological replicates were averaged for Vero cells. Keratinocytes were compared to fibroblasts by two-tailed paired $T$ test, $P = 0.001$, df = 5. Keratinocytes compared to Vero by a two-tailed unpaired student's $T$ test, $P = 0.004 - 0.025$ for df = 5 to 2. **E** Average $IC_{50}$ values for each donor and cell type (*** $P < 0.001$, * $P = 0.022$, linear mixed model accounting for dependent structure of multiple donors); analysis included five (keratinocytes) or three (fibroblast) averaged biological replicates in six donors. Five biological replicates were averaged for Vero cells. **F** Dose-response curve of acyclovir in Vero cells (purple), keratinocytes (blue), and fibroblasts (green). Solid lines represent averages, shaded region represents standard deviation. **G** Representative live cell images of keratinocytes, fibroblasts, and Vero cells infected with GFP-expressing HSV-1 and then treated with acyclovir at the specified doses were selected from at least three biological replicates (Scale bar 500 μm). **C**, **D**, **F**, **E** Plotted values are mean ± standard deviation. Source data provided for (**B**–**F**).

($\Delta RFU$) = 8753, $P < 0.001$), while the tdTomato signal was not affected ($\Delta RFU = 201.9$, $P = 0.8828$) by viral infection (Fig. 2D). We applied these optimal infection conditions to subsequent experiments.

We used two different infection methods, the submerged and ALI skin tissues, to emulate infection routes for primary and recurrent viral encounters. The submerged culture contains an undifferentiated single epithelium layer and was infected apically, resembling HSV primary infection at initial viral exposure. The ALI model included a stratified epidermis infected basolaterally, mimicking viral reactivation from the dermis. In the submerged model, apical infection of HSV-1 resulted in GFP expression mainly detected in keratinocytes above the tdTomato-positive dermal fibroblasts (Fig. 2E). In ALI cultures infected from the basolateral route of infection, there was colocalization of GFP and tdTomato signals, suggesting infection mainly occurred in the fibroblasts (Fig. 2F). H&E and IHC staining demonstrated that infected submerged cultures had a disrupted epithelial monolayer, with the

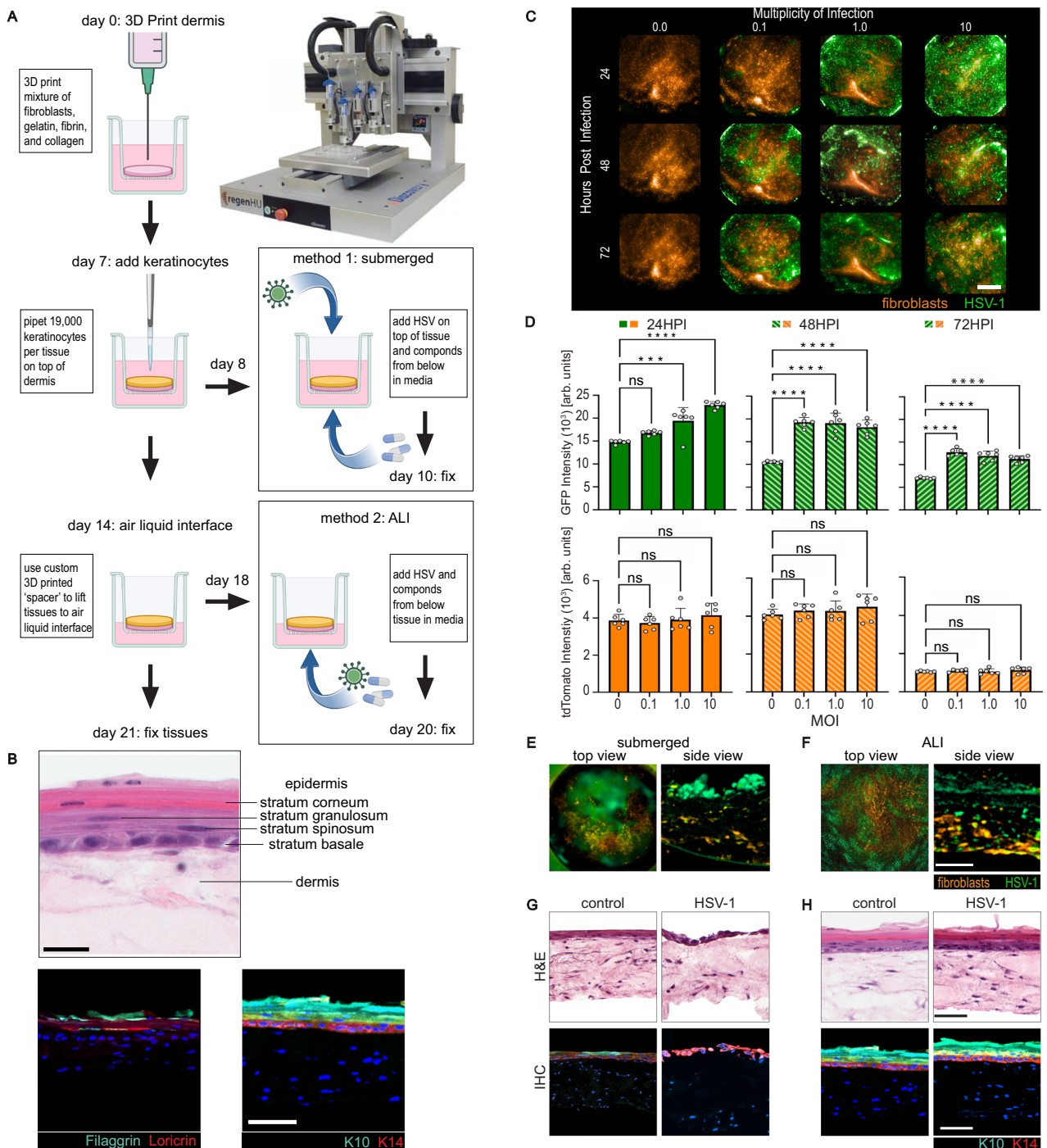

**Fig. 2 | 3D bioprinted HSE assay development and validation. A** Dermis equivalents were 3D printed onto the apical side of transwell inserts using the RegenHU 3D Discovery bioprinter (image courtesy of RegenHU). Keratinocytes were pipetted onto the apical surface of the dermis. In the submerged model, the tissues were infected at the apical surface. In the ALI model, tissues were brought to ALI and then infected at the basolateral surface (Created in BioRender. Ellison, S. (2025) https://BioRender.com/cege2fm). **B** H&E and IHC representative images of differentiated ALI tissues (*N* = 2). Loricrin (red) and filaggrin (cyan) expressed in stratum granulosum and stratum corneum, respectively, and K10 (cyan) and K14 (red) identify keratinocytes in the suprabasal and basal layer of the epidermis, respectively, and Hoechst nuclear stain in blue, (scale bars 20 µm and 50 µm). **C** Submerged tissues were infected at various MOI and then imaged at specified times. Fibroblasts express tdTomato (orange) while infected cells express GFP (green) (scale bar 1 mm). **D** GFP and tdTomato signal at each MOI and timepoint (*N* = 6 biological replicates, *** *P* < 0.001, **** *P* < 0.0001 by ordinary one-way ANOVA). We noted a decrease in GFP and tdTomato signal over time due to photobleaching. To correct for this reduction in fluorescence signals, we controlled each tested MOI to the proper uninfected control. Data is represented by mean values with error bars ± SD. [24 h: $P_1$ = 0.0002, $P_{10}$ < 0.001] [48 and 72hrs all *P* < 0.001]. Representative maximum projection images (*N* = 2) of infected tissues from the top and side view display viral infection (green) and the healthy fibroblasts (red) in the submerged (**E**) and ALI (**F**) tissues. H&E and IHC staining of uninfected or infected submerged (**G**) or ALI (**H**) models. (scale bars 50 µm). Source Data provided for (**D**).

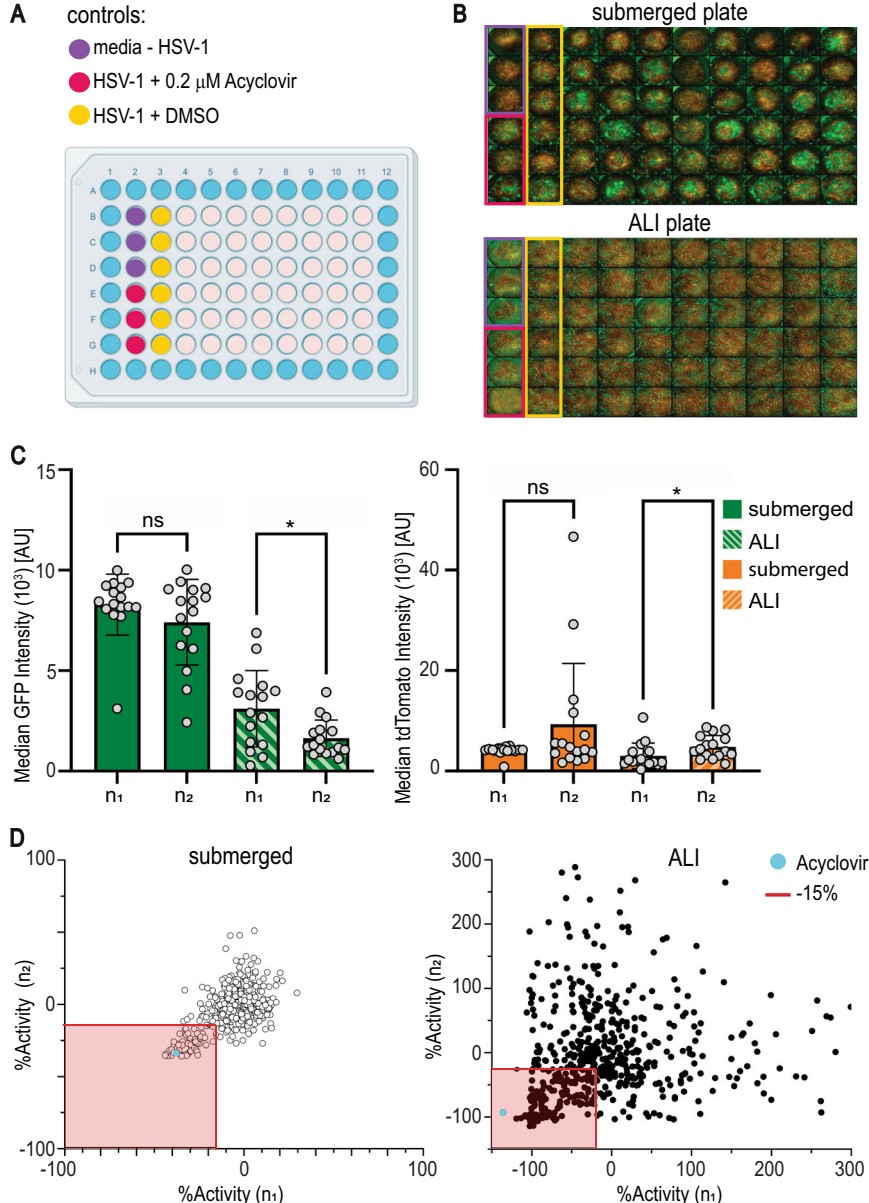

**Fig. 3 | Primary screen of compound library in 3D bioprinted assay platform.**
**A** Plate layout for compound screen. **B** Representative images from primary screen of submerged and ALI models. Controls are highlighted in boxes with colors corresponding to part (**A**). **C** Two-tailed Wilcoxon two-sample analysis of GFP intensity (left) and tdTomato intensity (right) of the median of the six negative control wells (HSV-1 + DMSO) across $N = 16$ plates per replicate, * $P = 0.023$ and 0.012 for GFP and tdTomato, respectively). Bars represent average with error bars as standard deviation. **D** Correlation plots of the two replicates for the primary screen of compounds. The red shaded box represents compounds that were selected as hits for re-testing if they reduced GFP expression (%Activity) by at least 15% in either replicate or model (left submerged, right ALI). Acyclovir was included in the collection and denoted by a cyan circle. Source Data provided for (**C**) and (**D**).

keratinocyte layer losing integrity (Fig. 2G), while infected ALI tissues maintained stratified epidermal morphology with proper K10 and K14 expression (Fig. 2H). Since the submerged and ALI infection models preferentially targeted keratinocytes and fibroblasts, respectively, we used both models for anti-HSV drug screening.

To validate our screening assay pharmacologically, we tested acyclovir in dose-response in submerged and ALI models (Supplementary Fig. 5). The $IC_{50}$ of acyclovir was > 9-fold higher in the submerged tissues ($IC_{50} = 0.36\,\mu M$), which infect mainly keratinocytes, compared to the ALI skin tissues ($IC_{50} < 0.04\,\mu M$), which infect predominantly fibroblasts. As expected, acyclovir inhibited HSV-1 gene expression with minimal cytotoxic effects on the fibroblasts. These data confirmed that our 3D HSE model could capture cell-type specific antiviral properties of acyclovir.

## Implementation of a primary drug screen using 3D bioprinted human skin tissue equivalents

We implemented a screen of 738 compounds with broad targets and a wide range of mechanisms of action (MOA) (Supplementary Data 1) utilizing our 3D bioprinted HSE assay platform. A primary screen tested the compound library in both the submerged and ALI infection models at 10 μM compound concentration in duplicate, using 3,840 bioprinted HSE tissues. Each plate contained control wells: negative control (HSV-1 + DMSO), inhibitor control (media -HSV-1), and a known compound (HSV-1 + ACV (0.2μM)) (Fig. 3A). We used the maximum projection of each well to determine the total fluorescence of GFP (HSV-1 activity) and tdTomato (fibroblast viability) and normalized the data to the control wells in the corresponding test plate as described in Materials and Methods (Fig. 3B, Supplementary Fig. 6, Supplementary Data 2).

To confirm the reproducibility of our assay, we compared the median GFP and tdTomato fluorescence signal from the negative control wells between the two assay repeats ($n_1$ vs $n_2$) in both the submerged and ALI models. The difference in the GFP or tdTomato intensity of the control wells was not significant in submerged models, indicating that our HSV-1 gene expression and fibroblast viability were consistent among all 32 plates (Fig. 3C, $P_{GFP} = 0.3045$, $P_{tdTomato} = 0.5641$). The difference in the GFP and tdTomato intensity of the control wells in the ALI plates was significant (Fig. 3C, $P_{GFP} = 0.0234$, $P_{tdTomato} = 0.0121$), emphasizing the need to include and normalize to controls on each plate to correct for any variability.

To determine if our assay is amenable to high-throughput screening (HTS), we calculated the median $Z'$-factor, a measure of the robustness of the assay. A score that is $Z' > 0.5$ denotes a robust assay window for screening, $0.5 > Z' > 0$ indicates a marginal assay window and that the screen needs replicates, and a $Z' < 0$ means that the assay window is not robust enough for screening. In the submerged model, the $Z'$ was 0.57, and for the ALI model, the $Z'$ was 0.13, indicating that the submerged assay is highly robust while the ALI assay is moderately robust, demonstrating that our 3D bioprinted assay platform was amenable to HTS (Supplementary Fig. 7)[39]. We identified 'hits' as compounds that inhibited GFP expression (HSV-1 activity) by at least 15% without causing more than 50% reduction in tdTomato fluorescence (fibroblast viability) in either screen mode (Fig. 3D), selecting 106 compounds from the primary screen.

## Secondary drug testing with 3D bioprinted human skin equivalents

Next, we performed a secondary screen of the 106 selected 'hits' in dose-response from 10 μM to 0.04 μM in the submerged and ALI models (Supplementary Data 3). We found that 46% of compounds in the submerged and 70% compounds in the ALI model reduced GFP expression (%Activity) by at least 50% at the maximum concentration tested without reducing tdTomato signal by 50% (%Viability) (Fig. 4A). Tested compounds were categorized by concentration-response curve (CRC) class (Fig. 4B)[39], according to their potency ($IC_{50}$) and efficacy (maximum percent activity). Compounds were considered candidate antivirals if they had CRC classes of −1.n or −2.n for HSV-1 inhibition and CRC class of 4 for fibroblast viability in at least one infection model. Forty-one candidate antivirals were identified and grouped by MOA (Fig. 4C). The MOA clustering indicated that HDAC inhibitors (8%) and protease inhibitors (8%) were more prevalent candidates in the submerged model while DNA polymerase inhibitors (13%) were more prevalent antivirals in the ALI model (Supplementary Fig. 8). Inhibitors of proteasome, DNA polymerase, Casein Kinase 2, Exportin-1, GBF-1, and Ribonucleotide-Reductase were effective in both models. Of the 41 candidate antivirals, 23 are known or experimental HSV treatments (Fig. 4D, E, Supplementary Table 2), including five 'ciclovir' drugs, acyclovir, ganciclovir, penciclovir, valaciclovir, and valganciclovir, the most common class of HSV antivirals[40–60], as well as two helicase-primase inhibitors: amenamevir and pritelivir[42,46,57]. Thus, our two-step assay strategy can identify HSV antivirals in an unbiased screen.

Based on their CRC classification, antiviral potency, and low fibroblast toxicity, we selected 11 of the 41 compounds as top-candidate antivirals for further testing. These included two known anti-herpes drugs, amenamevir and pritelivir[42,46,57], and four that have been reported to have anti-herpes activity: gemcitabine[52], lanatoside-C[54], niclosamide[48], and SNX-2112[58]. We re-tested these compounds in triplicate to assess potency and efficacy (Table 1, Fig. 5A, Supplementary Data 4). The $IC_{50}$ and maximum inhibition (Fig. 5B, C) indicate that several compounds were significantly more potent in the ALI model than in the submerged model, which is consistent with the implication of cell-type-specific antiviral properties. Three compounds, fimepinostat, LDC4297, and VLX1570, showed significant differences in potency

between the two models (Fig. 5B). We next plotted the viability of the tdTomato fibroblasts (Fig. 5D) and calculated the $CC_{50}$ (Table 1). While none of the compounds were cytotoxic in the ALI model as detected by tdTomato intensity, lanatoside C, SNX-2112, and VLX1570 exhibited some cytotoxicity in submerged models at higher concentrations. Further investigation using H&E and IHC staining in ALI infected HSE 48 h after drug treatment at 3.3 μM, we found normal appearance of epithelium for most candidate compounds tested, but flouroemetine, lanatoside-C, niclosamide, and SNX-2112 displayed visible signs of epithelium disruption, indicative of putative cytotoxicity to the keratinocytes at the higher dosage (Supplementary Fig. 9).

## Assessment of the candidate antiviral activities in donor-derived keratinocytes and fibroblasts

Differences in the $IC_{50}$ values between the submerged and ALI models suggested that, like acyclovir, some drugs may exhibit cell-type specific antiviral properties. We next investigated the 11 top candidate antivirals in our 2D monocultures of donor-derived primary keratinocytes and fibroblasts in dose-response to further investigate cell-type-specific effects, in comparison to acyclovir (Fig. 6A). Donors #3, #4, and #5 from Fig. 1 were used to test each top candidate antiviral. According to the $IC_{50}$ and maximum inhibition, all 11 candidate antivirals were more potent than acyclovir in keratinocytes (7 to >884-fold), with $IC_{50}$ values ranging from 0.08 to 9.9 μM (Fig. 6A–C, and Supplementary Fig. 10). Five candidate antivirals, amenamevir, pritelivir, gemcitabine, lanatoside-C, and SNX-2112, surpassed acyclovir antiviral potency in fibroblasts as well, with $IC_{50}$ values ranging from 0.05 to 0.15 μM (Fig. 6A, B, Supplementary Fig. 11).

Antiviral potency differed significantly between keratinocytes and fibroblasts for 8 of the 12 compounds in donor-matched skin cells (Fig. 6B, C, Supplementary Fig. 12). However, the differences in candidate antivirals were less than in acyclovir (< 27 fold versus >220 fold, respectively). Two notable exceptions were verdinexor and fimepinostat, which inhibited viral GFP expression by at least 50% in keratinocytes but failed to inhibit viral GFP expression in fibroblasts reliably. We also detected significant differences in antiviral responses between donors #3 and #4 ($P < 0.001$) and between donors #5 and #4 ($P = 0.02$) when considering potencies for all 11 compounds in keratinocytes, but not fibroblasts (Supplementary Figs. 13–15, Supplementary Table 3).

We next examined compound toxicity on keratinocytes and fibroblasts in the absence of viral infection (Fig. 6D, Supplementary Figs. 10, 11). Keratinocytes were generally more sensitive to compound cytotoxicity than fibroblasts. Amenamevir and pritelivir, which target viral helicase/primase, showed a highly effective antiviral dosage ($IC_{50}$ through $IC_{80}$), which was at >100-fold range lower than the $CC_{50}$, yielding a selectivity index of >250 in keratinocytes and >400 in fibroblasts (Fig. 6D, Supplementary Table 4). On the other hand, gemcitabine, SNX-2112, and lanatoside-C, which inhibit host pathways involving DNA replication, have high selectivity indexes ( > 750–1200) only in the fibroblasts and showed toxicity in keratinocytes.

To confirm candidate antivirals were effective against HSV-2 as well, we conducted a plaque reduction assay. Primary keratinocytes in 6-well plates were infected with either HSV-1 K26 or HSV-2 186 and treated with multiple doses of each candidate antiviral. All 11 candidate antivirals inhibited HSV-1 and HSV-2 plaque formation at similar doses (Supplementary Fig. 16).

## Comparison of candidate antivirals in 2D monoculture versus 3D bioprinted human skin equivalents

We compared the potency, toxicity, and selectivity index of the top 11 candidate compounds in our four models: 2D keratinocytes, 2D fibroblasts, 3D submerged, and 3D ALI (Fig. 7A, B, Supplementary Table 5). The impact of cell type and 2D:3D model type on the antiviral potency was compound-specific. Acyclovir was most significantly

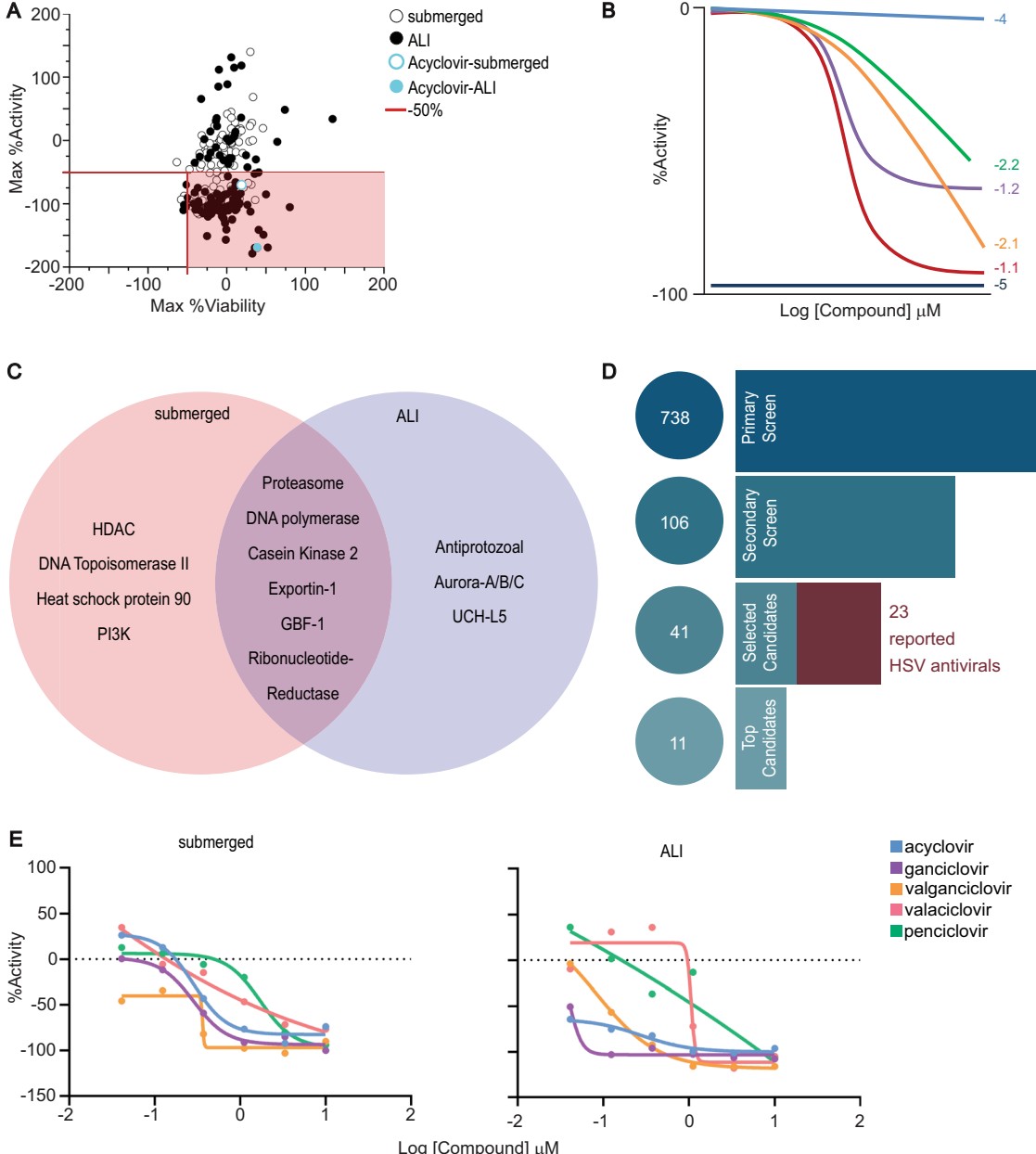

**Fig. 4 | Dose response of candidate antivirals in 3D bioprinted assay platform.**
**A** Correlation plot of Max %Activity (maximum reduction in GFP) vs. Max %Viability
(maximum reduction in tdTomato) of 106 'hits' tested in dose response. Top candidate antivirals (50% or greater reduction in GFP) that did not kill over 50% of tdTomato transduced fibroblasts are identified by the red shaded box. **B** Schematic illustrating %Activity dose response profiles of different Concentration-Response Curve classes (CRC). **C** Venn diagram showing divergent and coinciding targets for 41 top candidate antivirals in both submerged and ALI models. **D** Schematic of compounds selection from 738 compounds in the primary screen to 106 'hits' tested in dose-response to 41 selected candidates and 11 top candidates selected to move forward. Of the 41 selected candidates, 23 are current or experimental HSV treatments. **E** Dose response curves of candidate antivirals in "ciclovir" family, known to treat HSV-1, in submerged and ALI models ($N = 1$). Source Data provided for (**A**) and (**E**).

affected by cell type and the 2D:3D environment, whereas lanatoside C and SNX-2112 were unaffected (Supplementary Table 6). Submerged tissues and keratinocytes displayed more similar potencies than ALI tissues and fibroblasts, most likely due to the undifferentiated monolayer of keratinocytes infected in the submerged model (Fig. 7A). Combining cell type differences between keratinocytes and fibroblasts and the 2D:3D growth environment, we showed that amenamevir, pritelivir, fluoroemetine, gemcitabine, lanatoside C, and SNX-2112 exhibited less than a 3-fold difference in average potency between models (Fig. 7C–E); all remaining candidate antivirals were >3-fold more potent in the 3D model. Keratinocyte 2D monocultures were

more susceptible to cytotoxic effects of candidate antivirals than 2D monocultures of fibroblasts or the 3D models, which identify toxicity through a loss of fluorescently labeled fibroblasts (Fig. 7B). All candidate compounds were selective (selectivity index >10) in the ALI model, and 9 of the 11 were selective in submerged models. While 8 of the 11 candidate antivirals were selective in 2D fibroblasts, only the two helicase-primase inhibitors were selective in 2D keratinocytes. Antiviral potency and toxicity can significantly differ between cell type and 2D:3D model tested, emphasizing the importance of applying physiologically relevant models to assess drug responses early in development.

**Table 1 | Pharmacologic profile of top candidate antivirals**

| Candidate antiviral | Primary mechanism of action | Target | Submerged | | | | ALI | | | |
|---|---|---|---|---|---|---|---|---|---|---|
| | | | Curve Class | Max Inhibition (%) | $IC_{50}$ ($\mu$M) | $CC_{50}$ ($\mu$M) | Curve Class | Max Inhibition (%) | $IC_{50}$ ($\mu$M) | $CC_{50}$ ($\mu$M) |
| Amenamevir (AMN) | Helicase primase inhibitor | Viral | −1.1 | 85.31 | 0.27 | >10 | −1.1 | 112.36 | 0.16 | >10 |
| Pritelivir (PRT) | Helicase primase inhibitor | Viral | −1.1 | 94.64 | 0.50 | >10 | −1.2 | 101.90 | 0.21 | >10 |
| Fimepinostat (FMP) | PI3K/HDAC inhibitor | Host | −1.1 | 87.43 | 1.48 | >10 | −5 | 88.60 | <0.04 | >10 |
| Fluoroemetine (FLR) | Unknown | Host | −1.1 | 96.50 | 0.15 | >10 | −1.1 | 111.83 | 0.22 | >10 |
| Gemcitabine (GMC) | Ribonucleotide Reductase | Host | −1.1 | 99.84 | 0.19 | >10 | −1.2 | 101.78 | 0.16 | >10 |
| Lanatoside C (LNT) | Autophagy inducer | Host | −1.1 | 106.86 | 0.09 | 2.49 | −1.2 | 99.49 | 0.08 | >10 |
| LDC4297 (LDC) | CDK inhibitor | Host | −2.1 | 94.82 | 0.68 | >10 | −1.1 | 99.87 | 0.11 | >10 |
| Niclosamide (NCD) | Multi-functional | Host | −1.1 | 97.13 | 0.39 | >10 | −1.1 | 98.67 | 0.11 | >10 |
| SNX-2112 (SNX) | HSP90 inhibitor | Host | −1.2 | 101.78 | 0.05 | 1.14 | −5 | 107.14 | <0.04 | >10 |
| Verdinexor (VRD) | Exportin Antagonist | Host | −1.3 | 68.57 | 0.48 | > 10 | −1.1 | 107.81 | 0.17 | >10 |
| VLX1570 (VLX) | Protease deubiquitinase inhibitor | Host | −2.5 | 55.49 | 6.67 | 8.41 | −1.1 | 101.22 | 0.16 | >10 |

Top candidate antivirals were selected from 41 compounds identified as potent and effective at doses below their respective $CC_{50}$ in the primary and secondary screens. We re-tested these top candidate antivirals in triplicate and reported the Curve Class, Max Inhibition, $IC_{50}$, and $CC_{50}$.

### Generation of 3D bioprinted human skin equivalents using adult donor-derived skin cells

Human neonatal skin cells are readily available commercially, but the ability to use adult donor-derived cells would increase the translational impact of our drug discovery assay platform. We bioprinted the dermis with neonatal human dermal fibroblasts, then seeded the dermis with keratinocytes from Donor #3 as proof of concept to generate donor-specific HSE. H&E staining of submerged and ALI HSE showed donor-derived keratinocytes successfully developed into a stratified epidermis in the adult-derived ALI model with proper expression of K10 and K14 determined by IHC (Fig. 8A, B). Most of the 11 candidate antivirals displayed similar $IC_{50}$ values in neonatal-derived and adult donor-derived HSE (Fig. 8C, D, Supplementary Table 7, Supplementary Data 5). Thus, our assay platform could use easily accessible neonatal cells to initially screen large numbers of compounds inexpensively and then incorporate adult human skin cells for further preclinical evaluation to enhance drug testing in diverse genetic backgrounds.

## Discussion

Our study highlights critical variations in the susceptibility of different cell types to HSV manifestation and the effectiveness of acyclovir, a widely used antiviral drug for treating herpes diseases. We have demonstrated that acyclovir exhibits substantially lower potency in suppressing HSV infection in keratinocytes than in fibroblasts or Vero cells. Keratinocytes are the primary cell type encountered during HSV reactivation and peripheral infection. Human skin biopsies obtained during genital herpes ulceration or asymptomatic reactivation have shown that viral gene expression primarily occurs in keratinocytes within the skin epidermis. Innervating nerve endings, where reactivating viruses are released, reach to the basal keratinocytes, whereas CD8 + T cells localize contiguously to the nerve ending and directly contact basal keratinocytes[28,33]. The earlier initiation of viral gene expression and faster viral replication and growth rate observed in these cells further confirm that keratinocytes are the more physiologically relevant targets for antiviral drug screening and preclinical testing. This underscores an important limitation of current antiviral testing approaches, which utilize Vero cells and fibroblasts monolayer cultures, and has significant implications for HSV drug development strategies. We have established a high-throughput assay platform that leverages 3D bioprinting to rapidly generate over 6500 human skin

equivalents (HSEs) consisting of keratinocytes and fibroblasts, enabling the identification of potent HSV antiviral compounds. The 3D bioprinted HSE recapitulates the in vivo skin architecture, including the epidermis, dermis, and dermal-epidermal junction[30]. A GFP-expressing HSV-1 strain and tdTomato-expressing fibroblasts allowed our platform to rapidly screen for potency, efficacy, and cytotoxicity using 738 distinct compounds. We identified 41 potential antiviral candidates, 23 of which are known or experimental HSV antivirals, demonstrating the ability of our assay to identify drugs active against HSV in an unbiased screen. Finally, the successful integration of donor-derived adult skin cells into our platform to test antiviral responses further enhances the translational relevance of our preclinical drug screening system.

We demonstrated that the potency and efficacy of antiviral compounds could vary significantly between cell types and 2D versus 3D environments. The varied drug potency and efficacy between donor-matched keratinocytes and fibroblasts is best illustrated by acyclovir. This mainstay drug suppresses HSV lesion formation and reduces the duration of ulcer lesions but has limited efficacy in preventing subclinical shedding and transmission to sexual partners[12,15]. The $IC_{50}$ of acyclovir in donor-derived keratinocytes was over twice the peak of clinical serum levels following three times daily treatment with 1000 mg valacyclovir[35], a dose substantially higher than 1000 mg once per day used for standard suppressive therapy with valacyclovir[61]. Furthermore, mathematical modeling predicts that current treatment regimens result in ineffective doses of acyclovir for over 40% of the day[62]. The higher $IC_{50}$ of acyclovir in keratinocytes that we describe here, combined with the relatively short half-life of acyclovir[63], likely contributes to the episodic cycle of asymptomatic shedding during suppressive therapy. Together, the increased HSV gene expression and reduced acyclovir potency in keratinocytes might explain why acyclovir therapy cannot eliminate subclinical shedding, the associated HSV transmission, and the increased risk of HIV acquisition[15,16,64].

Our multi-model drug screening and evaluation strategy also yielded insight into HSV antiviral drug development. Our data showed antivirals like amenamevir and pritelivir[42,57], which inhibit HSV helicase-primase, were highly selective in tested 2D keratinocytes, 2D fibroblasts, 3D submerged, and 3D ALI models, and exhibited similar potencies against HSV-1 and HSV-2 strains, in agreement with published work[42,57]. Drugs that exhibited less than a twofold difference in

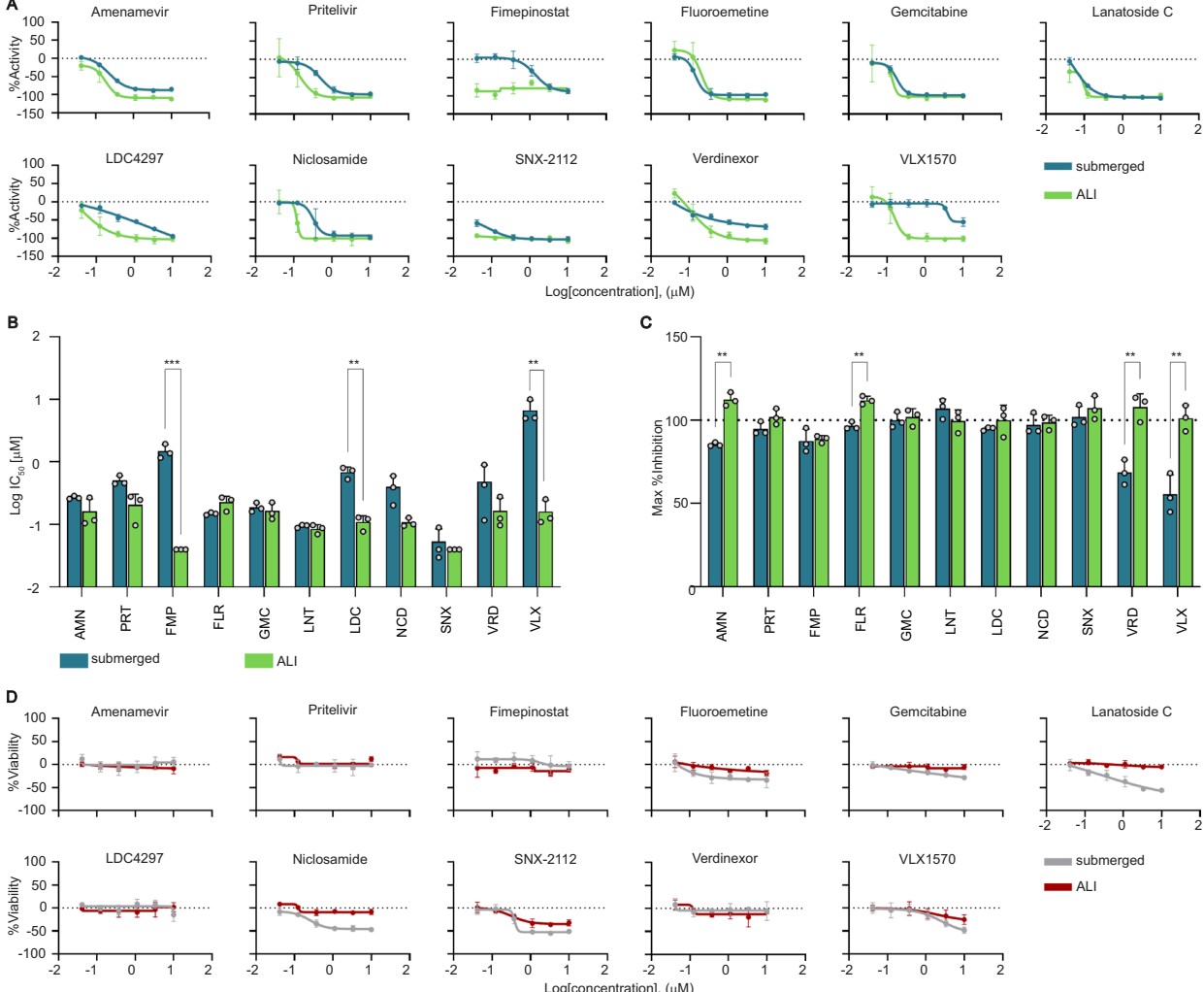

**Fig. 5 | Top candidate antiviral potency, efficacy, and cytotoxicity in 3D bioprinted HSE. A** Average dose-response curves for the 11 top candidate antivirals for submerged (blue) and ALI (green) models. **B** IC$_{50}$ values for each top candidate antiviral compared between submerged (blue) and ALI (green) models. **C** Maximum inhibition for each top candidate antiviral compared between submerged (blue) and ALI (green) models. Statistical significance was determined Wald tests based on linear models with multiple comparison adjustment (*** $P < 0.001$, ** $P < 0.01$, * $P < 0.05$) for (**B**) and (**C**). Refer to Supplementary Table 6 for specific $P$ values. **D** CC$_{50}$ dose response curves for 11 top candidate antivirals. All data is plotted as the average of three biological replicates ($N = 3$), error bars represent standard deviation. Source Data provided for all panels.

potency for all models included pritelivir, amenamevir, gemcitabine, SNX-2112, and lanatoside C. Interestingly, the first four of these drugs directly target viral DNA synthesis, while lanatoside C functions by preventing viral genetic material from entering the nucleus[42,54,57,58,65]. While our data suggested that targeting viral helicase-primase machinery was effective, drugs targeting host cell replication machinery for DNA synthesis could be effective across multiple cellular environments, provided that toxicity is not observed in complex models.

It is worth noting that our 3D bioprinted HSE screening detects cellular toxicity from tdTomato-expressing fibroblasts in a high-throughput, large-scale screening setting. Additionally, it enables the examination of stratified epidermal keratinocytes for pathomorphological effects on selected candidate compounds by histology and immunohistochemistry staining. Our data suggest that helicase-primase inhibitors may be safe in multiple cell types, as well as in both 2D and 3D environments. Compounds such as SNX-2112, Lanatoside C, Fluoroemetine, and Niclosamide may cause toxicity and disrupt epithelial integrity.

The reduced efficacy of acyclovir in keratinocytes could reflect inherent biological differences in drug metabolism and viral replication dynamics between cell types. Acyclovir is an acyclic purine nucleoside analog and functions as a prodrug. It is activated through phosphorylation by HSV thymidine kinase. Following this initial phosphorylation, two more phosphates are added to ACV-P through cellular kinase, resulting in ACV-triphosphate to terminate DNA chains and eliminate HSV. Thus, for acyclovir to inhibit viral DNA synthesis, it must be processed by cellular thymidine kinase in two of the three phosphorylation steps, which could introduce a source of cell-type-specific variability. In keratinocytes, HSV expression occurred more rapidly with a shorter lag time and faster kinetics than in Vero cells or donor-matched fibroblasts, suggesting their greater susceptibility to HSV infection and replication, which might be related to stem-like cellular nature of basal keratinocytes. While targeting viral DNA synthesis was generally effective in all models tested, acyclovir was substantially less potent in keratinocytes than fibroblasts, as demonstrated in 2D monolayer cultures and 3D bioprinted HSE. Future research is needed to uncover these mechanisms to improve antiviral strategies.

There are several aspects in which our engineered skin tissue models can be improved for higher physiological relevance to HSV infection. First, we could improve infection of stratified epithelium in

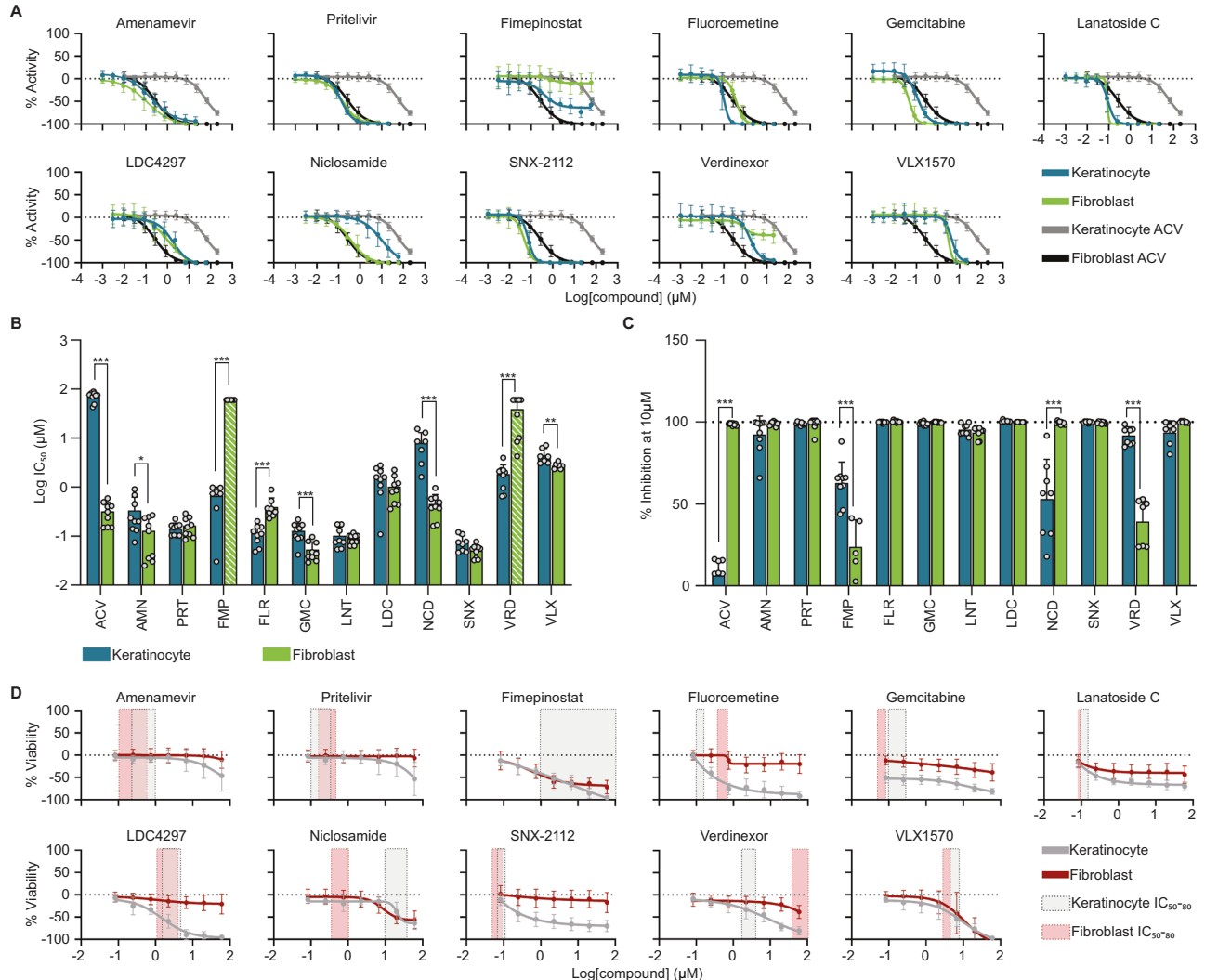

**Fig. 6 | Candidate antiviral potency, efficacy, and cytotoxicity in donor-derived primary keratinocyte and fibroblast monocultures. A** Dose-response curves for the 11 top candidate antivirals compared to acyclovir (ACV) (keratinocytes blue and gray, respectively; fibroblasts green and black, respectively). **B** IC$_{50}$ values for each top candidate antiviral compared between keratinocytes (blue) and fibroblasts (green). Striped bars (FMP, VRD) indicate candidate antivirals that failed to reduce GFP expression by at least 50% consistently. **C** Maximum inhibition for each top candidate antiviral is compared between keratinocytes (blue) and fibroblasts (green). Statistical significance was determined by Wald tests based on linear mixed models with multiple comparison adjustments accounting for the dependent structure of multiple donors (*** $P < 0.001$, ** $P < 0.01$, * $P < 0.05$) for (**B**) and (**C**). Refer to Supplementary Table 6 for specific $P$ values. **D** CC$_{50}$ dose-response curves for all twelve candidate antivirals compared to their respective IC$_{50}$ to IC$_{80}$ dose ranges. Keratinocyte data is from 20HPI (gray), while fibroblast data is from 48HPI (red). All data is plotted as the average of three replicates in three distinct donors ($N = 9$), error bars represent standard deviation. Source data provided for all panels.

the ALI culture through wounding at the apical surface to bypass the physical barrier of cornified epithelium. Second, our 3D models measure off-target drug toxicity using the tdTomato signal expressed by fibroblasts. Future iterations of this model could incorporate a third fluorophore into keratinocytes to measure toxicity, allowing for real-time tracking of compound toxicity to keratinocytes in the skin epidermis. Third, constructing HSV-2 GFP expressing recombinant virus similar to HSV-1 K26 would benefit more direct and rapid testing against HSV-2 infection. Other strategies of antiviral drug development against wildtype and clinical isolates of HSV could include modifying target cells to express GFP under a viral promoter.

Our assay platform can also be expanded by incorporating more donor-derived cells into the skin tissues. Incorporating broad genetic diversity for drug screening could improve clinical trials and patient outcomes, particularly when variable cellular factors significantly impact the efficacy of candidate drugs[66,67]. Our platform's ability to rapidly screen many different compounds could also be leveraged to screen combination therapy between a select group of compounds.

Combination therapy could be used to overcome cell-type-specific pharmaceutical properties or prevent the emergence of viral resistance, like the approach employed by antiretroviral therapy in treating HIV infection[30].

While this current study focuses on high-throughput antiviral drug discovery, the 3D human skin mimetics present an attractive model for deciphering virological and immunological features associated with HSV infection. Studies using an in vitro vascularized 3D human skin-on-chip platform for modeling human HSV infection have yielded pathophysiological insights that mimic HSV ulceration in humans, as well as elucidated the early events of cytokine-mediated innate immune responses, such as IL-8-dependent neutrophil chemotaxis and directional migration[29]. Our 3D HSE model would enable the interrogation of innate immune pathways in structural cells to understand interferon responses or other key cytokine/chemokine-mediated immune defenses against HSV. Future studies using donor-derived structure cells and matching immune cells would make it possible to delineate adaptive immune responses in a personalized setting.

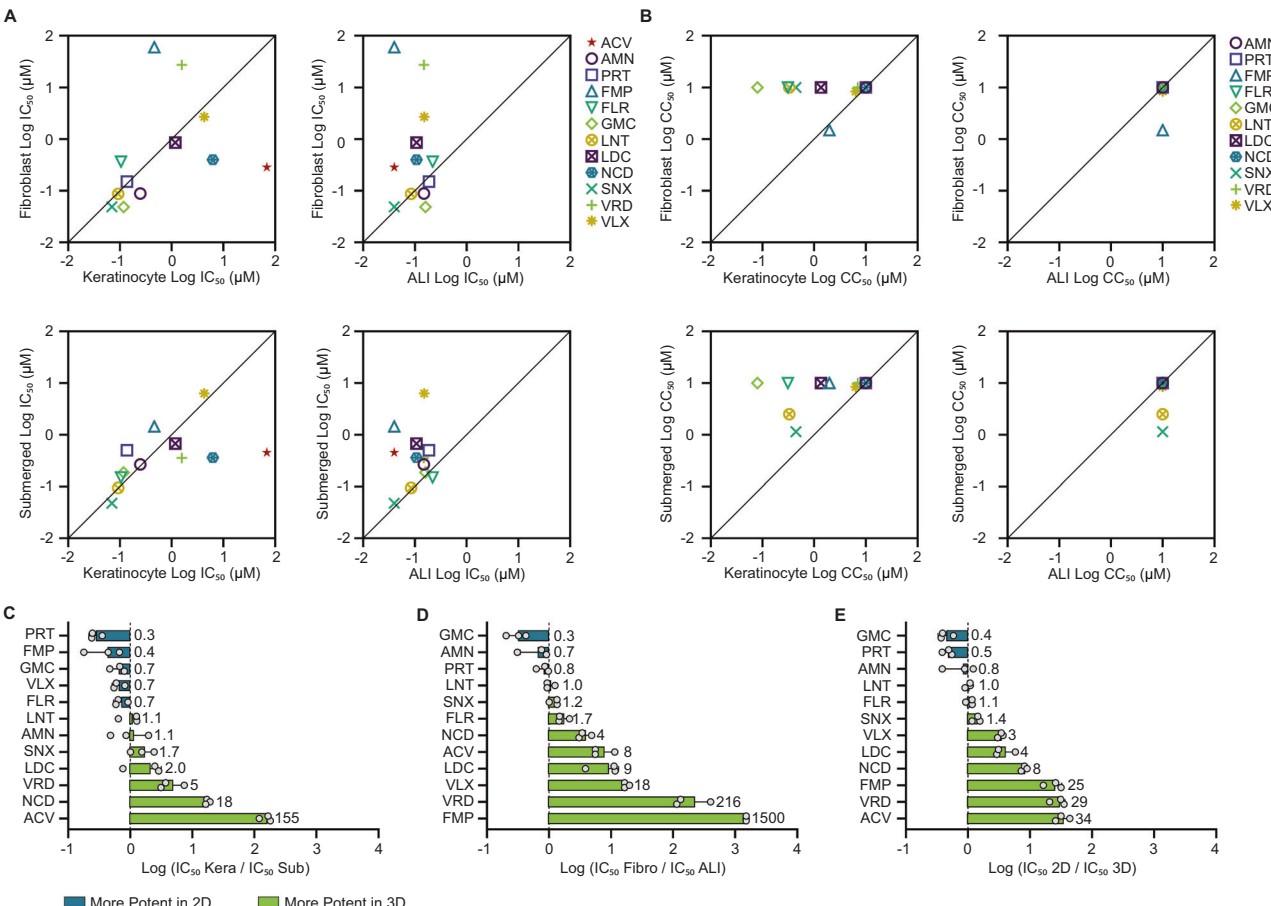

**Fig. 7 | Comparison of 3D and 2D models in testing antiviral candidates.**
**A** Pairwise comparisons of $IC_{50}$ values for candidate antivirals in the four models tested. **B** Pairwise comparisons of $CC_{50}$ values for candidate antivirals in the four models tested. **C** Fold change was determined by dividing the $IC_{50}$ value of each candidate antiviral in keratinocytes by the $IC_{50}$ of the same candidate antiviral in submerged models. **D** Fold change was determined by dividing the $IC_{50}$ value of each candidate antiviral in fibroblasts by the $IC_{50}$ of the same candidate antiviral in ALI models. **E** $IC_{50}$ values for each candidate antiviral were pooled (keratinocytes and fibroblasts, submerged and ALI), then $IC_{50}$ values for candidate antivirals in 2D were divided by $IC_{50}$ values in 3D. **C**, **D**, **E** Green bars indicate candidate antivirals that were more potent in 3D, while blue bars indicate candidate antivirals that are potent in 2D. Data for 2D ($N = 9$) and 3D ($N = 3$) is reported as averages, error bars are standard deviation. Source data provided for all panels.

In summary, we have established a 3D bioprinted HSE assay platform that can be used for drug screening to identify anti-viral compounds in a high-throughput format. We demonstrated that the potency and efficacy of candidate antivirals can vary significantly between cell types in both 2D cellular and 3D tissue cultures, with substantially lower potency observed for acyclovir in keratinocytes. Further, we identify that cell-type specific differences are largely absent in helicase-primase inhibitors, supporting the continued investigation of both this class of antivirals and antivirals targeting viral proteins. Finally, we establish that our assay platform can incorporate donor-derived cells, allowing early stages of drug discovery to include multiple donors with varied ages, sex, ancestry, and genetic backgrounds to improve success in downstream clinical trials.

## Methods
### Ethical statement
This study protocol involving human specimens was approved by the University of Washington Institutional Review Board committee (STUDY00004312). Written informed consent was collected from all participants. Skin punch biopsies were collected from HIV-seronegative healthy individuals ($n = 6$), either HSV seropositive ($n = 3$) or HSV seronegative ($n = 3$). Both male ($n = 1$) and female ($n = 5$) participants, aged from the 20 s to the 70 s, were included. Three-millimeter punch biopsies were obtained from normal skin of the inner

upper arm. Keratinocyte and dermal fibroblast cells were isolated from skin tissue, primarily cultured, and stored at −80 °C. Donor-specific keratinocyte and dermal fibroblast cells were matched for comparison.

### Study design
3D human skin equivalents (HSE) generation was performed in controlled experiments. Commercial neonatal fibroblasts and keratinocytes were expanded and frozen so that every experiment was performed at the same passage and using the same lot numbers. HSV-1 stocks were stored at −150 °C and only used once to avoid freeze-thaw cycles. Candidate antivirals were spotted into 96 well receivers from the same stocks for all studies. Spotted plates were stored at −80 °C and thawed on the day of use. Drug treatment was blinded in large scale screen. All tissues were fixed at 48 h post infection (HPI). Candidate antiviral wells were normalized to control wells, included on each plate. The primary screen was performed with duplicate biological repeats to facilitate rapid screening our large library of candidate compounds, with six replicates of control wells used per screening plate to monitor reproducibility. The final 11 candidate antivirals were tested in dose response with triplicate biological replicates following determination of a moderate Z-score, indicating more repeats were necessarily to accurately measure compound activity. All submerged and ALI plates were processed using a circular mask to remove false fluorescent signal from the edge of the wells. We also applied a size-based background removal on

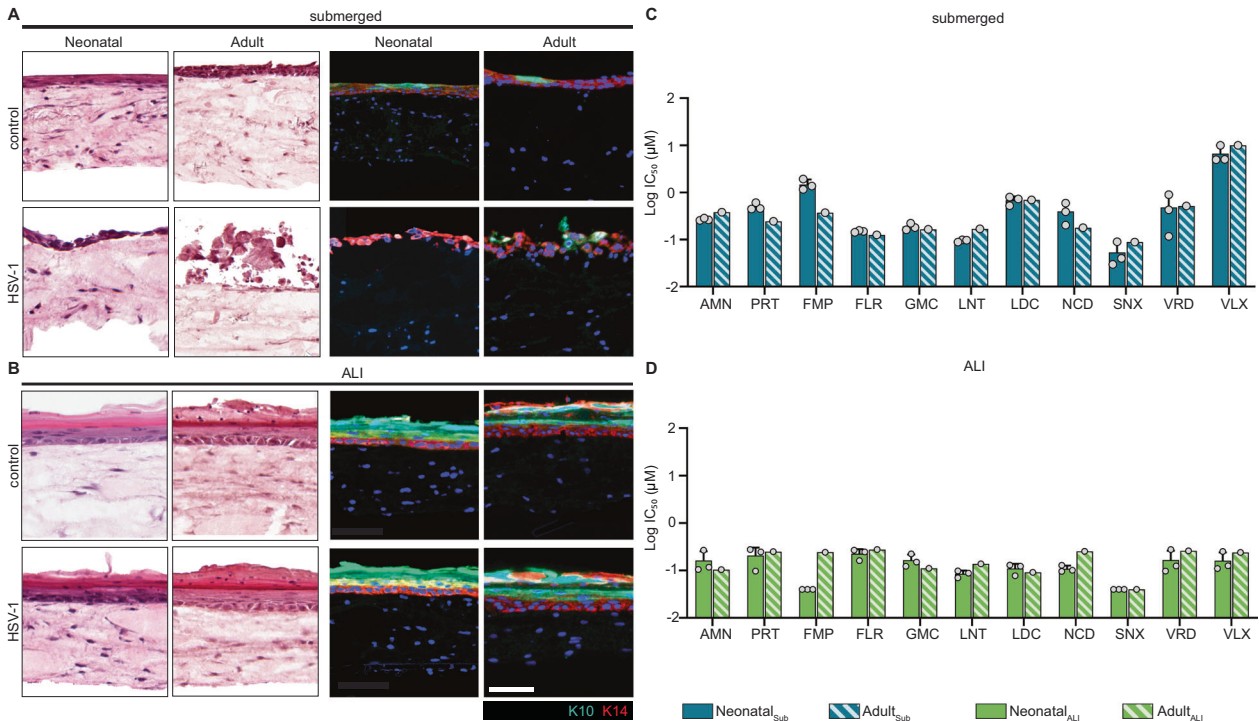

**Fig. 8 | Generation of 3D bioprinted HSE using adult donor-derived keratinocytes.** H&E and IHC staining of commercial neonatal or adult-derived bioprinted HSE in submerged (**A**) and ALI (**B**) models (scale bar 50 µm). Comparison of calculated $IC_{50}$ in submerged (**C**) and ALI (**D**) models for top candidate antivirals.

Neonatal data is plotted as an average of three biological replicates ($N = 3$), error bars are standard deviation. Adult data has an $N = 1$ and does not include error bars. Source Data provided for (**C**) and (**D**).

the ALI plates. All background removal steps were performed before the fluorescence of each well was measured and data was normalized. The research goal of 3D tissue studies was to develop a method to measure HSV-1 infection, candidate antiviral activity, and candidate antiviral toxicity in 3D printed HSE.

All 2D monoculture experiments were controlled laboratory experiments using low passage primary keratinocytes and fibroblasts. HSV-1 aliquots were used a single time without refreezing. All candidate antivirals were serially diluted in PBS from 10 mM stocks in DMSO and stored as aliquots at −20 °C. 2D monoculture experiments were not blinded, but data collection was automated to prevent bias. The primary endpoint for 2D monoculture experiments was determined by establishing the time of peak HSV-1 encoded GFP expression, providing a peak response to calculate IC50 values, while secondary endpoints included data collection and analysis at 20 and 48HPI or HPT. All candidate antivirals were tested in technical duplicate, and each cell type for each donor was tested in biological triplicate for each candidate antiviral to facilitate statistical analysis. A biological replicate is classified as an independent experiment using different assay preparations. Individual wells from all live cell imaging experiments were manually confirmed to be clear of autofluorescent debris, such as plastic particles, which would erroneously impact IncuCyte identification of fluorescent signal. Wells with autofluorescent debris were excluded from analysis. The primary research objective for 2D monoculture was to compare candidate antiviral potency and efficacy against 3D bioprinted HSE.

No commonly misidentified cell lines were used in this study. Primary cell types were confirmed through visual inspection of cellular morphology and the use of cell-type specific culture conditions.

## Commercial cell culture and transduction
Neonatal human dermal fibroblasts (HDF$_N$, Zen Bio DFN-F) were cultured in Dulbecco's Modified Eagle Medium (DMEM, Gibco 11965) supplemented with 10% Fetal Bovine Serum (FBS) and 1% penicillin

streptomycin. The fibroblasts were fluorescently labeled by transduction with lentiviral particles for fluorescent whole-cell labeling with tdTomato expression (Takara 0037VCT). A solution of polybrene and lentivirus at 20 multiplicities of infection (MOI) was added to the flask for 4 h at 37 °C, then virus solution was removed, and cells were expanded for 10 days, then frozen. Neonatal Normal Human Epithelial Keratinocytes (NHEK$_N$, ScienCell 2100) were cultured in KBM Gold basal medium (Lonza, Cat. # 00192151) supplemented with the KGM Gold Keratinocyte Growth Medium BulletKit (Lonza, Cat. #00192152). All cell types were incubated at 37 °C and 5% $CO_2$.

## Donor procured primary cell isolation and culture
Primary cell culture was completed using matched donor cells from 3 mm punch biopsies collected as previously described[68]. Briefly, punch biopsies were incubated in Dispase I (Sigma D4818) overnight at 4 °C, then the epidermis was mechanically separated from the dermis using tweezers. The epidermis was further digested with 0.05% trypsin (Gibco 25200-056) at 37 °C for 15 min, passed through a blunt needle (StemCell Technologies 28110) to generate a single cell suspension, and then seeded into collagen coated tissue culture vessels. The dermis was mechanically ripped into smaller pieces, gently air dried to adhere the dermis pieces to tissue culture vessels, then submerged in media. Fibroblasts emerged from adhered pieces of dermis over the next 96–120 h. All cell culture medium contained 25 mg streptomycin (Gibco 15140-122) and 25,000 units Penicillin (Gibco 15140-122). Keratinocytes were grown on collagen coated plates (Corning 35440) using DermaCult basal medium (Stem Cell Technologies 100-0501) with DermaCult keratinocyte expansion supplements (100-0502), 38 µg Hydrocortisone (Stem Cell Technologies 07925), and 125 µg Amphotericin B (Gibco 15290-019) added. The keratinocyte culture medium was supplemented with 10 µM of the Rho Kinase inhibitor Y-27632 dihydrochloride (Stem Cell 72304) for the first six days after collection[69]. Fibroblasts were grown on standard uncoated tissue culture vessels using Fibroblast Basal Medium 2 (PromoCell C-23220) with Fibroblast Growth Medium

2 supplement pack (PromoCell C-39320). All cells were maintained at 37 °C in a humidified incubator under 5% $CO_2$.

## Preparation of dermal base hydrogel

Gelatin powder (Sigma G1890, final concentration 0.045 mg/mL) was dissolved in a fibrinogen solution (Sigma F3879, final concentration 7.7 mg/mL) in PBS, at 37 °C. After complete dissolution of the gelatin, collagen I (Corning 354249, final concentration 4 mg/mL) was added, and the solution was mixed thoroughly. 10X phosphate-buffered saline (PBS, Invitrogen AM9624) was added to buffer the gelatin/fibrinogen/collagen solution. The mixture was kept at 37 °C and neutralized with 1 N NaOH immediately before adding the fibroblasts.

Fibroblasts at 70% confluence were dissociated, harvested from the flask, counted with a Countess II FL hemocytometer, and then centrifuged at 650 x $g$ for 4 min. The pellet was resuspended in 10 mL of the Dermal Base Hydrogel, at 2 million cells per mL, and mixed well by gentle pipetting. The solution was loaded into a 10 mL syringe (REGENHU 900013152) and chilled in the fridge for 5 min. Finally, a 0.42 mm ID luer lock needle tip (Nordson 7018263) was placed on the syringe and the syringe was loaded onto the RegenHU 3D Discovery bioprinter.

## 3D printing the dermal bioink

A plunger-based bioprinting system was used to extrude the dermal base hydrogel solution in a cylindrical-crosshatched pattern made up of two 0.25 mm thick layers with a diameter of 4.2 mm directly on top of the membrane (8 μm pore-size PET) of an HTS Transwell 96-well Permeable Support Plate (transwell plate, Corning 3374). The extrusion pattern was designed with the RegenHU BioCAD Software. To minimize evaporation effects, the hydrogel was not printed in any edge wells. After bioprinting was complete, the transwell was placed on a receiver plate filled with 250 μL per well of serum-free DMEM supplemented with 5units/mL thrombin (Sigma T6884) at RT to allow the fibrinogen-fibrin conversion. After 1 h, media in the receiver plates was changed to DMEM medium supplemented with 10% FBS, 1% penicillin streptomycin, and 0.025 mg/mL aprotinin (Sigma A4529). The plates were incubated at 37 °C and 5% $CO_2$; media was replaced every 48 h.

## Addition of keratinocytes

Seven days after bioprinting the dermal base, keratinocytes (NHEK$_N$) at 70% confluence were dissociated, harvested from the flask, counted using a hemocytometer, pelleted at 650 x $g$ for 4 min, then resuspended in KGM medium at 0.4 million cells per mL. 50 μL of the keratinocyte suspension (20,000 cells) was pipetted on top of the 3D bioprinted dermal base. Media in the receiver plates was changed to 250 μL of epidermalization medium per well supplemented with 0.025 mg/mL aprotinin[30,31,36]. The plates were incubated at 37 °C and 5% $CO_2$; media was replaced every 48 h.

## Air-liquid interface

7 days after adding the keratinocytes, the ALI model tissues were lifted to air liquid interface (ALI) using a custom 3 mm high 3D printed adaptor made of SBS-format compliant material. The adaptor lifted the transwell insert, leaving only the bottom of the dermal base in contact with the media while the top of the tissues was exposed to air. The media in the receiver plate was replaced with 510 μL of cornification medium supplemented with 0.025 mg/mL aprotinin[30,31,36]. Plates were incubated at 37 °C and 5% $CO_2$; media was replaced every 48 h.

## HSV infection in 3D bioprinted HSE

All viral infection experiments utilized a GFP-expressing recombinant HSV-1 strain, K26[38]. To infect bioprinted HSE, HSV-1 stocks were thawed over ice for ~1 h then diluted in media for the submerged infection method or cornification media for the ALI infection method. The optimal MOI of HSV-1 for infection assays in the 3D bioprinted HSE was determined by diluting HSV-1 in cell growth media to 0.1, 1.0, and

10 MOI, then adding the virus onto the apical side of the submerged tissues one day after seeding keratinocytes. Z-stacks were captured at 4X magnification on an inverted microscope, and a maximum projection was used to quantitate GFP and tdTomato total fluorescence signal of each well at 24, 48, and 72 h post infection (HPI). In the subsequent antiviral screening experiments, we used the HSV-1 at 0.1 MOI and fixed tissues at 48HPI.

We implemented two methods of infecting the 3D bioprinted HSE for antiviral screening. In the submerged method, the virus and the antivirals were added to the tissues one day after seeding the keratinocytes to the dermal base. The tissues were infected on the apical surface, but compounds were added to the media on the basal side of the tissues. In the ALI model, the virus and compounds were both added to the media on the basal side of the tissue four days after lifting to ALI to mimic viral reactivation.

## Compound plate preparation

For the screen, compounds were spotted into a 96-well receiver plates with the drug solutions at the appropriate stock concentrations in DMSO and frozen at −80 °C until ready for use. The plate map used for the screen is shown in Fig. 3A, with controls included on each plate including an inhibitor control (media -HSV-1), a negative control (HSV-1 + DMSO), and a known compound (HSV-1 + 0.2 μM acyclovir).

## Histology

Tissues were fixed in 4% paraformaldehyde for 24 h, then soaked in 30% sucrose for 24 h at 4 °C. The tissues were cut out from the transwell plate with a scalpel, then embedded in Scigen Tissue Plus O.C.T. Compound (Fisher Scientific 23-730-571) at −20 °C. The embedded tissues were sliced into 10 μm sections with a CryoStar NX50 and then mounted on positively charged slides. H&E staining was performed on the ThermoFisher Gemini stainer using the predefined H&E protocol. Using the Bond Fully Automated IHC Staining System, tissues were stained for primary antibodies Keratin 10: (1:2,000, ABCAM, ab234313, lots 1054991-1 and 1069014-1), Keratin 14: (1:500, ABCAM, ab119695, lot 1001149-31), filaggrin: (1:5,000, ABCAM, ab221155, lots 1029031-2, GR3456492-1, and GR3380631-6), and rabbit anti-human loricrin: (1:1,500, ABCAM, ab198994, lot 1063759-10) with Akoya Opal secondaries (1:150) in 690 (Fisher Scientific, NC1605064) and 520 (Fisher Scientific, NC1601877) respectively, and Hoechst nuclear stain (1:1000, ThermoFisher 33342). Slide images were captured at 10X magnification using the Leica Aperio Versa 200.

## Imaging and data normalization of 3D printed tissues

Tissues were fixed in 4% Paraformaldehyde for 24 h before washing with PBS. Fluorescent images were taken with a Molecular Devices ImageXpress High Content Reader using a 4X objective and excitation and emission wavelengths of 475–650 nm for GFP and 544-570 nm for tdTomato. Z-stacks of the images were taken at 25 μm step size and a maximum Z-projections were made for each well using the MetaXpress Image Analysis.

The fluorescent signal for each tissue was measured using the MetaXpress Image Analysis software. To remove the fluorescent signal of the well edges, a circular mask of ~4 mm in diameter was created and only signal within the 'circle mask' was measured. For the ALI tissues an additional step was included to decrease background autofluorescence signal; a size-based inclusion threshold of 14.8 μm or more was used to measure 'cell-sized' objects. (Supplementary Fig. 3) For both models, the GFP and tdTomato fluorescence intensity from the z-stack maximum projection was measured for each well. The intensity values were then normalized to the high and low signal controls using Eq. (1):

$$\% \, activity = 100x\frac{c-n}{n-i} \tag{1}$$

where $c$ is the candidate antiviral fluorescence intensity value, $n$ is the median signal from the negative controls (HSV-1 + DMSO), and $i$ is the median signal of the inhibition control (media -HSV-1) for the GFP signal and 0 for the tdTomato signal.

### Analysis of 3D model applicability for high-throughput screening

The reproducibility of results obtained using both the submerged and ALI 3D models was determined by Z-score, which was calculated using Eq. (2) [39]:

$$Z' = \frac{\left(AVG_{max} - \frac{3SD_{max}}{\sqrt{n}}\right) - \left(AVG_{min} - \frac{3SD_{min}}{\sqrt{n}}\right)}{AVG_{max} - AVG_{min}} \qquad (2)$$

where $AVG_{max}$=negative control (HSV-1 + DMSO) and $AVG_{min}$=inhibitor control (media -HSV-1), $SD_{max}$ and $SD_{min}$ is the standard deviation of the negative and inhibitor control respectively, and $n$ is the number of wells in each control group[39].

### 2D monoculture candidate antiviral testing

Primary keratinocytes were seeded in collagen coated 96 well plates (Revvity 6005810) and used at passage number six or less. Primary fibroblasts from the same keratinocyte donors were seeded in standard 96 well plates (Revvity 6005182) and used at passage number eight or less. 2D monocultures were infected with K26 HSV-1 using a MOI of 1.0 in serum free medium. Cells were infected at time zero and incubated on a rocker at 37 °C and 5% $CO_2$ for one hour prior to drug administration. Plates were imaged every two hours using a 10X objective on an IncuCyte S3/SX1. All images collected were analyzed using the IncuCyte software version 2021B (Sartorius). Infected wells treated with the 0.2% DMSO control were used to determine positive fluorescent signal while uninfected wells were used to determine background fluorescence for automated analysis. GFP expression is displayed as integrated intensity, defined as the total increase in fluorescent intensity as a function of the total area of detectable fluorescence.

Cytotoxicity was determined in the absence of HSV infection by treating cells with specified doses of candidate antivirals and using a combination of membrane permeable NucSpot Live 488 dye (Biotium 40081) and membrane impermeable NucSpot 594 dye (Biotium 41037). The cytotoxicity dose response curve was determined at 20 h post treatment (HPT) for keratinocytes and 48 HPT for primary fibroblasts to align with IC$_{50}$ timepoints.

### Plaque reduction assay

Six-well plates (Corning 3516) were coated with 0.04 mg/mL collagen I (Corning 354236) diluted in filter-sterilized 0.1% acetic acid (VWR UN2789) for 1 h, then seeded with primary keratinocytes. 95% confluent keratinocyte cultures were infected the next day using 40 plaque-forming units of either HSV-1 K26 or HSV-2 186. Cells were infected at time zero and incubated on a rocker at 37 °C and 5% $CO_2$ for one hour prior to drug administration. DermaCult was prepared as described above, then supplemented with 1% methylcellulose and candidate antivirals at the required concentrations. Excess virus was removed after one hour and replaced with overlay media containing candidate antivirals. Plaques were allowed to develop for 72 h at 37 °C and 5% $CO_2$, then cells were fixed for 30 min in 4% paraformaldehyde (Boster Bio AR1068) and stained with crystal violet (J.T. Baker F906-03) before manual counting.

### IncuCyte IC$_{50}$ calculations

All IC$_{50}$ values were determined by GraphPad Prism 9 using integrated fluorescent intensity of GFP measured by the IncuCyte software. IC$_{50}$ values were calculated at 20, 36, or 48 HPI as specified. Uninfected wells served as the negative control and were used as the minimum signal (0%), while drug-free infected wells were used as the positive control and represent the maximum signal (100%). IC$_{50}$ was calculated using a four parameter least squares nonlinear regression comparing the drug concentration to normalized response.

### Statistical analysis

Percent activity and percent viability in the 3D HSE was determined by normalizing the candidate antivirals to the control wells. Statistical analysis was completed using Student's $t$ test, Wilcoxon two-sample test, or linear mixed model as described in figure legends. All tests are two-sided. Holm's method was used in multiple comparison adjustment. A $p$ value < 0.05 was considered statistically significant.

Outliers in 2D monolayer studies were determined as values at least two logs higher than the mean. GFP fluorescence was normalized in 2D studies by subtracting fluorescence in mock infected wells and dividing by fluorescence in control untreated infected wells. Viability was determined in 2D studies by subtracting the total dead cells from the total cells to determine total live cells, then normalized by dividing the number of live cells in each experimental sample by the number of live cells in the 0.2% DMSO control at that timepoint. Statistical analysis for data sets involving multiple donors were analyzed by linear mixed models to account for donor-to-donor variability. Comparisons between submerged and ALI models were completed using linear models as a single cell source was used for all 3D tissues. Fold change calculations with 2D and 3D models used geometric means to account for the large differences between values.

### Reporting summary

Further information on research design is available in the Nature Portfolio Reporting Summary linked to this article.

## Data availability

Source data are provided with this paper for the main text and the supplementary materials.

## Code availability

The analysis R code is deposited in https://github.com/youyifong/drug_discovery_HSV/.

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

## Acknowledgements

This work was made possible by funding from the National Institutes of Allergy and Infectious Disease of the National Institutes of Health AI143773 (J.Z.) and T32 AI07140 (I.H.) and the National Center for Advancing Translational Sciences of the National Institutes of Health TR003208 (J.Z.). This research was supported in part by the Intramural Research Program of the National Institutes of Health (NIH) (M. F.). The contributions of the NIH author(s) made as part of their official duties as NIH federal employees are in compliance with agency policy requirements and are considered Works of the United States Government. However, the findings and conclusions presented in this paper are those of the author(s) and do not necessarily reflect the views of the NIH or the U.S. Department of Health and Human Services.

## Author contributions

S.T.E. designed, performed, and analyzed the 3D bioprinted human skin equivalent studies and wrote and reviewed the manuscript. I.H. designed, performed, and analyzed the 2D studies and wrote and reviewed the manuscript. K.D. and P.D. developed the 3D bioprinted human skin equivalent protocol. S.F. designed and manufactured the custom 3D printed lifters. Z.I. prepared the compound plates for the 3D bioprinted human skin equivalent studies. M.S. performed data analysis on 3D bioprinted human skin equivalent studies. A.J. and W.O. designed, performed, and analyzed the 2D monoculture experiments. L.C. discussed the data and reviewed the manuscript. A.W. and C.J. oversaw clinic operation and donor sample collection and reviewed the manuscript. Y.F. performed the data and statistical analysis. M.F. conceived, designed, and supervised the 3D bioprinted human skin equivalent studies, wrote and reviewed the manuscript, and obtained funding support. J.Z. conceived, designed, supervised, and analyzed the study, wrote and reviewed the manuscript, and obtained funding support.

## Competing interests

The authors declare no competing interests.
