## [Transparent Peer Review file · Nature Communications]

Limitations of acyclovir and identification of potent HSV antivirals using 3D bioprinted human skin equivalents

Corresponding Author: Dr Jia Zhu

Version 0:

Reviewer comments:

Reviewer #1

(Remarks to the Author)

Authors demonstrated that acyclovir is significantly less effective in skin-derived keratinocytes compared to donor-matched fibroblasts and compared to Vero cells (typically cell lines used for drug screening). They claimed to produce a 3D skin bioprinted platform for the screening of a library of 738 compounds. Authors also claim that this platform allows the integration of patient-derived cells which also includes genetic variability in drug development. Authors also claimed that the potency and efficacy of antiviral compounds could vary significantly between cell types and 2D versus 3D environments. This work is a very elegant approach which combines drug screening and personalized medicine. However, this referee has several concerns that need to be addressed:

1. Assessment of HSV Susceptibility and Acyclovir Antiviral Potency Across Different Cell Types (2D):
 - 1.1. It seems that there are not differences in the origin of 6 donors (some seropositives and other not) when they are treated with acyclovir.
 - 1.2. Acyclovir exhibited significantly reduced potency in keratinocytes compared to fibroblasts across all six donor. Could authors provide more details related to this effect? Is it because toxicity?
2. Development of HSV Infection Assays on 3D Bioprinted Human Skin Equivalents.
 - 2.1. Authors print dermis but the deposition of keratinocytes is performed manually/pipetting. If this platform will be used for drug screening, why this deposition is not performed automatically?
 - 2.2. H&E and IHC images of differentiated ALI skin tissues. Epidermis is not well formed. It is showed the generation of basal and corneum but there are missing layers. No basal membrane is showed.
 - 2.3. Methodology (submerge and ALI) needs to improve:
 - 2.3.1. since the infection is limited to it (for example, submerge implies only keratinocytes infection and ALI implies only fibroblast infection) and it does not mimic a real infection.
 - 2.3.2 It would be good if skin models last more time to follow-up the infection and treatment.
 - 2.3.3. It would be good if you can measure toxicity of both cell lines (keratinocytes and fibroblasts). Indeed, I would focus on the toxicity of keratinocytes rather fibroblasts.
 - 2.3.4. H&E and IHC staining demonstrated that infected submerged cultures had a disrupted epithelial monolayer. Authors do not show any control to compare.
 3. Implementation of a Primary Drug Screen using 3D Bioprinted Human Skin Tissue Equivalents.
 - 3.1. How many replicates are used for each compound (738)? How do authors manage to manual/pipetting keratinocytes in the 3,840 bioprinted HSE tissues?
 4. Secondary Drug Testing with 3D Bioprinted Human Skin Equivalents (performed a secondary screen of the 106 selected 'hits')
 - 4.1. What is the reason: "more potent in the ALI than the submerged model in 10 of the 11 290 compounds tested (Table 1)".
 5. Assessment of the Candidate Antiviral Activities in Donor-Derived Keratinocytes and Fibroblasts 2D.
 - 5.1. Could authors provide more information related to the origin of donor-derived primary keratinocytes and fibroblasts?
 6. Generation of 3D Bioprinted Human Skin Equivalents using Adult Donor-Derived Keratinocytes.
 - 6.1. Why do authors use neonatal fibroblasts and not only keratinocytes from adult donor-derived?
 - 6.2 Origin of 3 donors?
 - 6.3. Figure 8, epidermis is not well formed. It is showed the generation of basal and corneum but there are missing layers.
 7. Over 6,500 human skin equivalents (HSE). Is there any way to analyze/characterize that HSE are well formed in terms of epidermis and dermis?
 8. Lines 417-418: Authors do not demonstrate the generation of dermal-epidermal junction (DEJ)
 9. Do authors consider repeated doses and treatment during time in the infected HSE models?

Reviewer #2

(Remarks to the Author)

Ellison et al. presented a novel platform utilizing 3D bio-printed human skin equivalents for discovering antivirals against HSV infection. The study begins with evaluating acyclovir efficacy in donor-derived keratinocytes and fibroblasts, followed by validation in both submerged and air-liquid interface (ALI)-cultured skin equivalents incorporating neonatal fibroblasts and keratinocytes. Subsequently, the authors screened a drug library in the 3D skin model and compared antiviral efficacy between 2D monolayer cultures and 3D skin equivalents. While the study is new for HSV research, there is many repetitive work in testing/validating antiviral compounds and the advantages of the 3D skin equivalent compared to existing models (e.g., skin organoids) are not clear. The current manuscript's structural organization is difficult for the reader to follow the key findings and their significance.

Specific comments:

1. To establish the 3D bioprinted skin model as a robust platform for studying viral infections, a comprehensive mapping of the HSV life-cycle within the model is essential. The current data only provided limited description of intracellular HSV replication through fluorescence detection over a short time period. The authors should investigate both intracellular replication and extracellular virus release over an extended period (e.g., 1 hour to 7 days post-infection). Moreover, are the viral particles preferentially secreted from the apical or basolateral side ?
2. The manuscript lacks an evaluation of the host response to HSV infection within the 3D bioprinted model. It would be interesting and valuable to assess whether the infection triggers innate immune pathways, such as interferon responses. Performing transcriptomic profiling (bulk RNA sequencing) could provide critical insights into the antiviral response activated in this model.
3. It's interesting to see the difference between submerged and ALI-cultured skin equivalents models. A big advantage of ALI-cultured skin model is the differentiated keratinocytes and stratification. Considering this physiological feature, it would be more relevant to evaluate topical antiviral treatments, rather than just drug supplementation in the basolateral medium.
4. The authors transduced tdTomato in the fibroblasts to monitor compound cytotoxicity. However, how to assess the cytotoxicity of compounds to the keratinocytes within 3D bioprinted skin equivalents?
5. Immune cells play a pivotal role in the skin's defense against viral pathogens. The current model lacks this important component. The authors should discuss this potential limitation.
6. In Figure 2D, why GFP and tdTomato signal both decreased at 71 HPI compared to 24 or 48 hpi ? The authors should explain whether this reduction is due to cell death? diminished reporter expression or other reason?
7. Fig.S5B, in ALI culture, even the lowest concentration of acyclovir demonstrated over 50% anti-HSV activity?
8. According to the description in the Methods section, antiviral drug was treated immediately after one hour of viral infection. However, in clinical settings, there is often a delay between infection onset and treatment initiation. Therefore, a delayed treatment to mimic the clinical scenario is important in testing antiviral drugs in the model.
9. The authors claimed that "H&E and IHC staining demonstrated that infected submerged cultures had a disrupted epithelial monolayer (Fig.2E)" The image in Figure 2E is not convincingly supported this claim.
10. The reported IC50 values for acyclovir in submerged and ALI-cultured models are below 1 μ M. For robust antiviral screening, it is standard to use comparable or lower concentrations to identify new potent candidates. However, the authors screened 738 compounds at a relatively high concentration of 10 μ M, why? This may lead to cytotoxic effects, particularly many of the tested compounds are anti-cancer agents. The rationale for selecting this concentration should be clarified.
11. At the beginning, the authors used neonatal human fibroblasts and neonatal human keratinocytes for developing 3D skin equivalents and testing antivirals. Then at the last, the authors claimed that to increase the translational impact, neonatal fibroblasts and donor-derived keratinocytes were used for developing 3D skin equivalents. The authors should justify why donor-matched fibroblasts and keratinocytes were not used from the start, as this would be more logical and translationally meaningful.
12. The authors repeatedly test antiviral compounds in 2D cells, and 3D skin equivalents. Instead, this reviewer suggest to focus on the most significant/promising anti-HSV candidates and explore their mechanisms of action within the 3D bioprinted skin model.

Version 1:

Reviewer comments:

Reviewer #1

(Remarks to the Author)

The authors have properly addressed my concerns and comments.

Reviewer #2

(Remarks to the Author)

The reviewer has no additional comments. The authors have addressed most of the previous concerns in the revised manuscript.

Dear Reviewers:

We appreciate your valuable feedback and constructive suggestions. Our manuscript, "Limitations of acyclovir and identification of potent HSV antivirals using 3D bioprinted human skin equivalents," has been revised to provide further information addressing all comments and concerns raised by the reviewers. Please see our text and experimental revisions, along with the point-by-point responses described below.

Reviewer #1 (Remarks to the Author):

Authors demonstrated that acyclovir is significantly less effective in skin-derived keratinocytes compared to donor-matched fibroblasts and compared to Vero cells (typically cell lines used for drug screening). They claimed to produce a 3D skin bioprinted platform for the screening of a library of 738 compounds. Authors also claims that this platform allows the integration of patient-derived cells which also includes genetic variability in drug development. Authors also claimed that the potency and efficacy of antiviral compounds could vary significantly between cell types and 2D versus 3D environments. This work is a very elegant approach which combines drug screening and personalized medicine. However, this referee has several concerns that need to be addressed:

1. Assessment of HSV Susceptibility and Acyclovir Antiviral Potency Across Different Cell Types (2D):

1.1. It seems that there are not differences in the origin of 6 donors (some seropositive and other not) when they are treated with acyclovir.

Response: We thank the reviewer for the comment, we collected skin punch biopsies (3 mm) from the inner arm, an HSV-unaffected area, from six participants: three were HSV+ seropositive and three were HSV-negative. Donor-specific keratinocytes and dermal fibroblasts were isolated and matched for comparison. No donor-specific response to acyclovir treatment was observed from the six donors. Instead, a dramatic distinction occurred in the antiviral effect of acyclovir between keratinocytes and fibroblasts. We added additional sentences in the text to highlight that, **lines 105 and 116-120**.

1.2. Acyclovir exhibited significantly reduced potency in keratinocytes compared to fibroblasts across all six donor. Could authors provide more details related to this effect? Is it because toxicity?

Response: We appreciate this question. The potential toxicity effect of acyclovir can impact its antiviral activity observed in keratinocytes and fibroblasts. We performed dose-response experiments to examine acyclovir cytotoxicity in all primary donor-derived keratinocytes and fibroblasts at doses ranging from 2 μM to 600 μM . The CC_{50} for acyclovir in both fibroblasts and keratinocytes is over 600 μM . We have included this new data in **fig. S3** and in the text, **lines 122-124**.

Acyclovir is not known to be toxic *in vitro* (1), and indeed, it did not show increased toxicity to human primary keratinocytes and fibroblasts in our hands. We suspect the

differences in acyclovir potency between cell types arise from how HSV and/or acyclovir interacts with specific cell types. Our virological data suggested that the significantly reduced acyclovir potency in keratinocytes compared to fibroblasts in all six donors might be due to the faster kinetics of HSV gene expression and DNA replication. HSV GFP expression peaked 28 hours earlier in keratinocytes than fibroblasts (**Fig. 1B**). We now showed that HSV GFP expression initiated significantly earlier and doubled significantly faster in keratinocytes than fibroblasts (**Fig. 1C & D**). We have combined the original **Fig. S2 into Figure 1** to highlight these points and added to the discussion **in lines 411-416**. Our unpublished work also yielded similar findings in neutralization assays, indicating that more HSV-specific antibodies were required to neutralize HSV-infected keratinocytes than HSV-infected fibroblasts. We are interested in pursuing this in future studies, as this difference could also apply to other relevant therapeutic strategies.

2. Development of HSV Infection Assays on 3D Bioprinted Human Skin Equivalents.

2.1. Authors print dermis but the deposition of keratinocytes is performed manually/pipetting. If this platform will be used for drug screening, why this deposition is not performed automatically?

Response: Automatic deposition of keratinocytes by bioprinting methods was found to be challenging. Bioprinting requires cells to be suspended in hydrogels for extrusion-based printing or an agitator for jetting cell-laden droplets-based dispensation, respectively. Regardless of the method, we observed that bioprinting of keratinocytes likely stressed these sensitive cells and interfered with their differentiation into stratified skin epidermis. We therefore decided to manually dispense the keratinocytes by pipetting to avoid unnecessary negative effects of shearing forces by the bioprinter. We agree with the reviewer that cell deposition by pipetting can be automated using standard laboratory automation liquid dispensers. However, for the scale of the screens described here, we found that it was more practical and efficient to do manual pipetting using a 12-channel multipipette, which still enabled us to efficiently implement the screen and lowered the number of expensive cells needed by minimizing dead volumes. For future larger-scale drug screening with these biofabricated skin models, automated keratinocyte dispensing will indeed be very advantageous, and we are currently exploring this venue with the NCATS Automation group.

2.2.H&E and IHC images of differentiated ALI skin tissues. Epidermis is not well formed. It is showed the generation of basal and corneum but there are missing layers. No basal membrane is showed.

Response: We appreciate the reviewer's comment and understand the concern. The epidermis in Figure 2B was originally shown at 10X and we have now reimaged the tissues at 40X, and labeled the different epidermal layers in the H&E image to **Fig. 2B, Lines 159-160**. In addition, we have added images of IHC staining for loricrin and filaggrin, which are markers of the stratum granulosum and stratum corneum, respectively, in the stratified epidermis (**Fig 2B**), **Lines 162-163 and 633-634**. Although we did not stain the basal membrane in this manuscript, we previously shown that the protocol used to make this 3D bioprinted human skin equivalent (HSE) includes a Collagen-VII basal membrane underneath basal keratinocytes [(5), Fig. 4]. We hope that these additional images,

together with previously published data from our lab, will show convincingly that the skin tissue at ALI has a well-stratified epidermis.

2.3.1. Since the infection is limited to the model used (for example, submerge implies only keratinocytes infection and ALI implies only fibroblast infection) this model does not mimic a real infection.

Response: The reviewer raises a valid point. Although the model is limited due to the lack of simultaneous infection in both layers, our models still accurately reflected *in vivo* human herpes infection. Our submerged model was designed to capture the importance of basal keratinocytes on HSV-1 infection *in vivo*, as demonstrated by previously published work. First, studies (2, 3) using human biopsy tissue obtained during genital herpes ulceration or asymptomatic reactivation have indicated that viral gene expression primarily occurred in keratinocytes within the skin epidermis. Second, reactivated viruses released by nerve endings infect basal keratinocytes at the dermal-epidermal junction (DEJ). And third, CD8+ T cells localized contiguously to the nerve ending and directly contacted basal keratinocytes at the DEJ. These facts indicate that basal keratinocytes are the primary targets for HSV infection. We added to the Discussion Lines 411-414)

In the ALI model, applying HSV directly to the cornified layer resulted in no infection, indicating that HSE has a barrier function against infection (data not shown). To yield substantial infection in stratified epithelium in ALI cultures, it would require punch or scarification for wounding to circumnavigate the cornified layers (4), and this is not practical to execute on the large scale used for our drug screening. Instead, we added HSV to the dermis because fibroblasts can be infected *in vivo*, albeit at a much lower abundance. We feel it's essential to incorporate a method that enables us to assess antiviral efficacy in fibroblasts within a physiologically relevant environment containing extracellular matrix. Specifically, the current HSV drug discovery practice utilizes 2D fibroblast cultures. Together, using both infection models, we identified 23 compounds that showed HSV antiviral activities in published experimental work or clinical trials, demonstrating the relevance and robustness of our screening assays.

2.3.2 It would be good if skin models last more time to follow-up the infection and treatment.

Response: We agree with the reviewer that a longer lifetime of the biofabricated skin models would be beneficial to model certain skin diseases. In our experience, 3D bioprinted HSE typically lasts for three weeks. In addition, HSV infects skin relatively rapidly, yielding a productive infection within 24 to 72 hours, depending on the route of infection, so we believe that the lifetime of these bioprinted skin tissue models is sufficient to investigate HSV infection *in vitro*. Long-term skin culture models would indeed be needed for more slowly growing human viruses, such as human papillomavirus.

2.3.3. It would be good if you can measure toxicity of both cell lines (keratinocytes and fibroblasts). Indeed, I would focus on the toxicity of keratinocytes rather fibroblasts.

Response: We agree with the reviewer that it would be desirable to measure toxicities in both keratinocytes and fibroblasts in the tissues, simultaneously. We attempted to utilize

retroviral vectors to transduce GFP or RFP genes into primary human keratinocytes; however, we found that these fluorescently labeled keratinocytes do not form stratified epithelia properly. We continue to explore alternative methods for measuring compound toxicity on the keratinocytes in the epidermis layers. For now, we have assessed the cytotoxic effects of the candidate compounds using H&E and IHC staining. We have added these new images to the supplementary materials in **fig. S9** and in text **lines 291-295**.

2.3.4. H&E and IHC staining demonstrated that infected submerged cultures had a disrupted epithelial monolayer. Authors do not show any control to compare.

Response: We appreciate the reviewer's comment. The comparison between control and HSV-infected submerged HSE using H&E and IHC staining is now included in **Fig. 2E-H**. We have also edited the manuscript to clarify our results, **lines 196-200**.

3. Implementation of a Primary Drug Screen using 3D Bioprinted Human Skin Tissue Equivalents.

3.1. How many replicates are used for each compound (738)? How do authors manage to manual/pipetting keratinocytes in the 3,840 bioprinted HSE tissues?

Response: We thank the reviewer for this question. The initial screen included two biological replicates; Each replicate tested 1,920 individual tissues, which were screened into two separate batches of ten 96-well plates. We utilized 96-well transwell plates, which are compatible with standard high-throughput screening equipment and bioprinters. We have developed protocols to bioprint the fibroblasts hydrogel to make the dermis in an automated extrusion mode directly on the top side of the transwell membrane inserts (protocol previously published in ref 5). Because these 96-well transwell plates utilize standard HTS plate format, we were able to use a 12-channel multipipette to rapidly apply keratinocytes, change media, and add virus or drugs.

4. Secondary Drug Testing with 3D Bioprinted Human Skin Equivalents(performed a secondary screen of the 106 selected 'hits)

4.1. What is the reason: "more potent in the ALI than the submerged model in 10 of the 11 290 compounds tested (Table 1)".

Response: We speculate that the difference in submerged versus ALI antiviral potency is due to the different cell types preferentially infected: keratinocytes in the submerged and fibroblasts in the ALI model. We have added a clarifying sentence to the manuscript, **lines 286-287**. This is consistent with the differences in antiviral potency seen for the compounds for the infection in the two cell types in 2D.

5. Assessment of the Candidate Antiviral Activities in Donor-Derived Keratinocytes and Fibroblasts 2D.

5.1. Could authors provide more information related to the origin of donor-derived primary keratinocytes and fibroblasts?

Response: Please see our response to **Question 1.1.**

6. Generation of 3D Bioprinted Human Skin Equivalents using Adult Donor-Derived Keratinocytes.

6.1. Why do authors use neonatal fibroblasts and only keratinocytes from adult donor-derived?

Response: We performed additional experiments using donor-matched fibroblasts and keratinocytes from Donor #3; however, we found that when fabricating 3D bioprinted skin tissues with donor-matched keratinocytes and fibroblasts, the dermis did not form properly. We suspect several reasons could contribute to this problem: 1) The donor-derived fibroblasts may require lower passages; 2) A higher concentration of fibroblasts may be required in the dermal bioink; 3) Adult donor-derived fibroblasts may be incompatible with retroviral transduction. We would like to investigate this issue further in the future. However, we successfully 3D bioprinted tissues using neonatal fibroblasts and adult donor-derived keratinocytes, a more physiologically relevant cell type for HSV infection, providing a proof of concept that adult donor-derived primary cells can be utilized in 3D bioprinted skin tissues.

6.2 Origin of 3 donors?

Response: We used cells from Donor #3 for these experiments, the same Donor #3 used in the monolayer experiments. We have clarified this in the manuscript: line **392**.

6.3. Figure 8, epidermis is not well formed. It is showed the generation of basal and corneum but there are missing layers.

Response: Please see the above Response 2.2 to address this concern.

7. Over 6,500 human skin equivalents (HSE). Is there any way to analyse/characterize that HSE are well formed in terms of epidermis an dermis?

Response: Before performing the high-throughput screen of over 6,500 HSE, we developed a standardized protocol that consistently displayed a proper epidermis through histology. For the high-throughput screen, we randomly selected control wells to perform histology and confirm that the HSE consistently displays a differentiated and stratified epidermis. We did not individually test each tissue, as this would prohibit a high-throughput screen, but statistical analysis of reproducibility as measured by Z-score indicated our platform was sufficiently reproducible (**lines 240-245, fig. S7**).

8. Lines 417-418: Authors do not demonstrate the generation of dermal-epidermal junction (DEJ)

Response: We appreciate the reviewer's comment and understand their concern. Although we did not stain the basement membrane in this manuscript, we characterized and validated extensively the 3D bioprinted human skin equivalent (HSE) in our prior publication, including characterization of basement membrane formation by Collagen-VII

expression underneath basal keratinocytes [(5), Fig. 4], and confirming the presence of the basement membrane and fully stratified epidermis using the same 3D bioprinted human skin equivalent protocol we have developed in our lab.

9. Do authors consider repeated doses and treatment during time in the infected HSE models?

Response: We agree with the reviewer that studying repeated doses would be an interesting future direction and hope to incorporate repeated doses and treatments in the human skin equivalent to mimic treatment in the clinic. This method was not possible due to the fixed time scale of the HSE function, and with the high-throughput drug screening, but it could be applied to the final selected compounds. Future iterations can add new cell types, variable infection conditions, and alterations to treatment dosing and regimens.

Reviewer #2 (Remarks to the Author):

1. To establish the 3D bioprinted skin model as a robust platform for studying viral infections, a comprehensive mapping of the HSV life-cycle within the model is essential. The current data only provided limited description of intracellular HSV replication through fluorescence detection over a short time period. The authors should investigate both intracellular replication and extracellular virus release over an extended period (e.g., 1 hour to 7 days post-infection). Moreover, are the viral particles preferentially secreted from the apical or basolateral side ?

Response: We appreciate the Reviewer's thoughtful comments and agree that the 3D human skin mimetics present an attractive opportunity to investigate HSV susceptibility and viral replication in stratified epithelium. While this current study focuses on high-throughput HSV drug discovery using 3D bioprinted HSE, our prior study carefully addressed these virological questions in a published work, in which we developed a human skin-on-chip platform for modeling human HSV infection in vitro (6).

In that study, we examined HSV infectivity at different stages of keratinocyte differentiation. We tested both HSV-1 and HSV-2 strains for infectivity by applying viruses before or after air-liquid interface, at Days -1, 1, 3, 5, 7, and 9 [(6), Fig. 2]. We observed that undifferentiated keratinocyte monolayers (Day -1, before air lifting) were the most susceptible to HSV-1 and HSV-2 infection among all the time points tested. Keratinocyte differentiation after airlifting dramatically reduced HSV infectivity. HSV infection was undetectable in stratified epithelium 5 days post-airlifting, when the cornified epidermal layer formed, suggesting the importance of barrier function in resisting HSV infection.

Furthermore, we investigated intracellular HSV replication in the basal keratinocyte layer of a fully developed skin-on-chip epidermis using a mechanical punch to facilitate virus entry [(6), Fig. 3]. HSV infection primarily occurred in keratinocytes along the epidermal rupture. The virus-infected cells formed multinucleation through cell-cell fusion and displayed nucleus enlargement and chromatin margination, the pathomorphological features commonly observed in ulcerative genital herpes lesions. HSV spreading to the distal regions occurred mainly in K14+K10- basal keratinocytes. Thus, our studies highlight the role of basal keratinocytes in the manifestation and dissemination of HSV

infection. They also confirm that the 3D human skin equivalent can be used as a physiologically relevant in vitro model for HSV infection.

2 and 5. The manuscript lacks an evaluation of the host response to HSV infection within the 3D bioprinted model. It would be interesting and valuable to assess whether the infection triggers innate immune pathways, such as interferon responses. Performing transcriptomic profiling (bulk RNA sequencing) could provide critical insights into the antiviral response activated in this model. Furthermore, immune cells play a pivotal role in the skin's defense against viral pathogens. The current model lacks this important component. The authors should discuss this potential limitation.

Response: We appreciate the reviewer's suggestions and agree that the 3D HSE is an excellent model for deciphering immune defense against viral infection. In our previous publication, we have dedicated three figures [(6), Fig. 4, 5, 6] to investigating these valuable questions and understanding the early innate immune responses to HSV infection. We successfully detected cytokine expression of IL-8, IL-6, and RANTES in our 3D skin-on-chip platform and monitored neutrophil recruitment and migration in real-time. We also demonstrated that neutrophil chemotaxis and directional migration were dependent on IL-8 production. We have included a paragraph in the discussion on the application of using the 3D human skin model for dissecting host immune responses.
Lines 492-500.

3. It's interesting to see the difference between submerged and ALI-cultured skin equivalents models. A big advantage of ALI-cultured skin model is the differentiated keratinocytes and stratification. Considering this physiological feature, it would be more relevant to evaluate topical antiviral treatments, rather than just drug supplementation in the basolateral medium.

Response: We agree with the Reviewer that an ALI-cultured fully differentiated human skin model would be an excellent preclinical platform for developing topical antiviral treatment. Currently, this is beyond the scope of this study, but we hope to investigate this in the future.

4. The authors transduced tdTomato in the fibroblasts to monitor compound cytotoxicity. However, how to assess the cytotoxicity of compounds to the keratinocytes within 3D bioprinted skin equivalents?

Response: Primary human keratinocytes are known to be challenging to modify via conventional methods. We attempted to utilize retroviral vectors to transduce GFP or RFP genes into primary human keratinocytes; however, these fluorescently labeled keratinocytes were unable to form a stratified epithelium. Here, we have examined tissue morphology to assess the toxicity of the selected drug to the skin utilizing H&E and IHC staining. We have added these new images to the supplemental material **fig. S9** and text in **lines 291-295 and 454-459**. We continue to explore alternative methods for measuring keratinocyte toxicity.

6. In Figure 2D, why GFP and tdTomato signal both decreased at 71 HPI compared to 24 or 48

hpi ? The authors should explain whether this reduction is due to cell death? diminished reporter expression or other reason?

Response: We appreciate the reviewer pointing this out. The reduction in fluorescent signals at 72 HPI, compared to 24 and 48 HPI, was most likely due to photobleaching resulting from repeated imaging over a prolonged period. We added a clarification in the manuscript: **lines 179-181**.

7. Fig.S5B, in ALI culture, even the lowest concentration of acyclovir demonstrated over 50% anti-HSV activity?

Response: This is an excellent question. Acyclovir's cell-type-specific antiviral potency might explain this discrepancy, since fibroblasts were the primary cell type targeted in the ALI model. Acyclovir exhibited a potency well below 1 μ M in fibroblasts and was about 200-fold more potent in the fibroblast monolayer cultures than the keratinocytes. We believe the high potency in ALI cultures is compounded by the higher antiviral activity of acyclovir in the fibroblasts.

8. According to the description in the Methods section, antiviral drug was treated immediately after one hour of viral infection. However, in clinical settings, there is often a delay between infection onset and treatment initiation. Therefore, a delayed treatment to mimic the clinical scenario is important in testing antiviral drugs in the model.

Response: Antiviral therapy with acyclovir is commonly used as both a suppressive treatment and prescribed at the onset of symptoms, meaning acyclovir can be present before, immediately after, or after a short delay to lesion formation. We have previously published on how acyclovir efficacy was impacted by treatment time using a more complex *in vitro* human skin-on-chip model [(6), Fig. 7]. We found that even a 24-hour delay in treatment would result in a significant loss of antiviral activity compared to treatment administered 24 hours prior or simultaneously. Acyclovir treatment administered at the time of HSV inoculation demonstrated equal antiviral activity as the prior treatment. HSV has a relatively rapid replication lifecycle in cultured keratinocytes and fibroblasts. Without robust immune responses in the HSE *in vitro*, HSV viral replication can progress rather rapidly in as little as 48-72 hours, resulting in widespread infection, at which point antivirals will not be screened accurately. Therefore, we choose to treat immediately with all the candidate compounds.

9. The authors claimed that "H&E and IHC staining demonstrated that infected submerged cultures had a disrupted epithelial monolayer (Fig.2E)" The image in Figure 2E is not convincingly supported this claim.

Response: Please refer to the Response 2.3.4

10. The reported IC50 values for acyclovir in submerged and ALI-cultured models are below 1 μ M. For robust antiviral screening, it is standard to use comparable or lower concentrations to identify new potent candidates. However, the authors screened 738 compounds at a relatively

high concentration of 10 μM , why? This may lead to cytotoxic effects, particularly many of the tested compounds are anti-cancer agents. The rationale for selecting this concentration should be clarified.

Response: We appreciate the reviewer's question and acknowledge that there are multiple approaches to best screen candidate drugs. Since we were testing a broad library of compounds with a wide range of mechanisms of action and using acyclovir as a positive control, we performed the primary screen of 738 compounds at 10 μM , a standard practice in drug screening. We selected compounds using a lenient cut-off of 'inhibiting GFP expression (HSV-1 activity) by at least 15% without causing more than 50% reduction in tdTomato fluorescence (fibroblast viability) in the primary screening. This helped us narrow down the list to 106 compounds. We then performed a secondary screening using dose-response analysis, ranging from 10 μM to 0.04 μM . This two-step screening strategy identified 41 candidate antivirals, 23 of which were known or experimental treatments, including all five 'ciclovir' drugs: Acyclovir, Ganciclovir, Penciclovir, Valaciclovir, and Valganciclovir, as well as two helicase-primase inhibitor drugs: Amenamevir and Pritelivir (Supplementary Table S2). Thus, we demonstrated our assays can identify HSV antivirals in an unbiased screen.

11. At the beginning, the authors used neonatal human fibroblasts and neonatal human keratinocytes for developing 3D skin equivalents and testing antivirals. Then at the last, the authors claimed that to increase the translational impact, neonatal fibroblasts and donor-derived keratinocytes were used for developing 3D skin equivalents. The authors should justify why donor-matched fibroblasts and keratinocytes were not used from the start, as this would be more logical and translationally meaningful.

Response: Due to the limited proliferation capacities of adult donor-derived primary human fibroblasts and keratinocytes, we were unable to generate large quantities of primary human keratinocytes and fibroblasts sufficient to complete both the primary and secondary screens of all 738 compounds. The screening assays required over 120 million keratinocytes and fibroblasts, while our adult donor-derived skin cells, obtained from 3 mm punch biopsies, yielded significantly fewer cells. To overcome this limitation and ensure reproducibility, we employed commercially available neonatal human keratinocytes and fibroblasts, which enables broader compound screening and facilitates the adoption of our HSE assay platform by others without access to primary cell donors. Neonatal human keratinocytes and fibroblasts support large-scale screening efforts, but we recommend using donor-derived cells whenever feasible in late-stage validation studies to reflect patient-specific biology.

12. The authors repeatedly test antiviral compounds in 2D cells, and 3D skin equivalents. Instead, this reviewer suggest to focus on the most significant/promising anti-HSV candidates and explore their mechanisms of action within the 3D bioprinted skin model.

Response: We appreciate the reviewer's comments and fully agree that elucidating the mechanisms of action of candidate antivirals is the necessary and important next step.

The primary goal of this study is to provide proof of concept that an *in vitro* human skin equivalent model can be effectively used to screen anti-herpesvirus drugs. To this end, we employed both 2D and 3D models to validate the antiviral activities, compare their performance, and explore the notable observation that antiviral potency and efficacy can differ significantly between keratinocytes and fibroblasts. Our model's ability to select known anti-HSV drugs from the larger screen in an unbiased manner strongly supports its utility and validates its relevance. Although several top candidate compounds have established mechanisms of action against HSV (**Fig. 4C and Table 1**), we are interested in identifying mechanisms of action that differ between keratinocytes and fibroblasts, as well as between 2D and 3D models. Specific host pathways may be more effective antiviral targets in certain cell types or model systems. We are seeking future funding opportunities to develop further the most promising HSV antiviral candidates.

References

1. Elion GB, Furman PA, Fyfe JA, de Miranda P, Beauchamp L, Schaeffer HJ. Selectivity of action of an antiherpetic agent, 9-(2-hydroxyethoxymethyl) guanine. *Proc Natl Acad Sci U S A*. 1977;74(12):5716-20. doi: 10.1073/pnas.74.12.5716. PubMed PMID: 202961; PubMed Central PMCID: PMC431864.
2. Zhu J, Koelle DM, Cao J, Vazquez J, Huang ML, Hladik F, et al. Virus-specific CD8+ T cells accumulate near sensory nerve endings in genital skin during subclinical HSV-2 reactivation. *J Exp Med*. 2007;204(3):595-603. Epub 20070226. doi: 10.1084/jem.20061792. PubMed PMID: 17325200; PubMed Central PMCID: PMC2137910.
3. Zhu J, Peng T, Johnston C, Phasouk K, Kask AS, Klock A, et al. Immune surveillance by CD8alphaalpha+ skin-resident T cells in human herpes virus infection. *Nature*. 2013;497(7450):494-7. Epub 20130508. doi: 10.1038/nature12110. PubMed PMID: 23657257; PubMed Central PMCID: PMC3663925.
4. De La Cruz NC, Mockel M, Wirtz L, Sunaoglu K, Malter W, Zinser M, et al. Ex Vivo Infection of Human Skin with Herpes Simplex Virus 1 Reveals Mechanical Wounds as Insufficient Entry Portals via the Skin Surface. *J Virol*. 2021;95(21):e0133821. Epub 20210811. doi: 10.1128/JVI.01338-21. PubMed PMID: 34379501; PubMed Central PMCID: PMC8513464.
5. Derr K, Zou J, Luo K, Song MJ, Sittampalam GS, Zhou C, et al. Fully Three-Dimensional Bioprinted Skin Equivalent Constructs with Validated Morphology and Barrier Function. *Tissue Eng Part C Methods*. 2019;25(6):334-43. doi: 10.1089/ten.TEC.2018.0318. PubMed PMID: 31007132; PubMed Central PMCID: PMC6589501.
6. Sun S, Jin L, Zheng Y, Zhu J. Modeling human HSV infection via a vascularized immune-competent skin-on-chip platform. *Nat Commun*. 2022;13(1):5481. Epub 20220919. doi: 10.1038/s41467-022-33114-1. PubMed PMID: 36123328; PubMed Central PMCID: PMC9485166.